# Fine-Grained Semantically Aligned Vision-Language Pre-Training

**Juncheng Li** [1,2*†]    **Xin He** [2†]    **Longhui Wei** [2†]    **Long Qian** [1]    **Linchao Zhu** [1]

**Lingxi Xie** [2]    **Yueting Zhuang** [1‡]    **Qi Tian** [2‡]    **Siliang Tang** [1‡]

[1] Zhejiang University, [2] Huawei Cloud
{junchengli, qianlong0926, zhulinchao, yzhuang, siliang}@zju.edu.cn
{hexin80, weilonghui1, tian.qi1}@huawei.com, 198808xc@gmail.com

## Abstract

Large-scale vision-language pre-training has shown impressive advances in a wide range of downstream tasks. Existing methods mainly model the cross-modal alignment by the similarity of the global representations of images and texts, or advanced cross-modal attention upon image and text features. However, they fail to explicitly learn the fine-grained semantic alignment between visual regions and textual phrases, as only global image-text alignment information is available. In this paper, we introduce **LOUPE**🔍, a fine-grained semantically a**L**igned visi**O**n-lang**U**age **P**r**E**-training framework, which learns fine-grained semantic alignment from the novel perspective of game-theoretic interactions. To efficiently compute the game-theoretic interactions, we further propose an uncertainty-aware neural Shapley interaction learning module. Experiments show that LOUPE achieves state-of-the-art performance on a variety of vision-language tasks. Furthermore, without any object-level human annotations and fine-tuning, LOUPE achieves competitive performance on object detection and visual grounding. More importantly, LOUPE opens a new promising direction of learning fine-grained semantics from large-scale raw image-text pairs. The repository of this work is at https://github.com/YYJMJC/LOUPE.

## 1 Introduction

Learning transferable cross-modal representations from large-scale vision-language pre-training has exhibited remarkable performance on a wide variety of downstream tasks. Most existing works can be classified into two categories: *dual-encoder* and *fusion-encoder*. The dual-encoder methods [17, 26, 35, 49] adopt two separate encoders to embed images and texts, and model the cross-modal alignment by the cosine similarity between the global features of images and texts. While such architecture is efficient for large-scale image-text retrieval by pre-computing image and text representations offline, they fail to model fine-grained semantic alignment between visual regions and textual phrases. On the other hand, the fusion-encoder methods [7, 19, 20, 25, 30, 34, 43, 42] attempt to use a single multi-modal encoder to jointly model the concatenated sequence of images and texts. These methods simulate soft alignment via advanced cross-modal attention [45]. However, they can only learn implicit alignment by end-to-end training, lacking explicit supervision to encourage semantic

---

*Work done when interning at Huawei Cloud.

†Equal Contribution.

‡Corresponding Authors.

alignment between visual regions and textual phrases. And the learned cross-modal attention matrices are often scattering and uninterpretable. Further, they are inefficient for retrieval since it requires jointly encoding every image-text pair during inference.

Learning fine-grained semantic alignment from image-text pre-training is crucial to many cross-modal reasoning tasks (*e.g., visual grounding* [51], *image captioning* [48]), but it is particularly challenging as the alignment information between visual regions and textual phrases is not available, posing fine-grained semantic alignment learning a weakly-supervised learning problem. In this paper, we address this problem while simultaneously maintaining high retrieval efficiency by proposing **LOUPE**🔍, a fine-grained semantically a**L**igned visi**O**n-lang**U**age **PrE**-training framework, from the novel perspective of game theory. We formulate input patch and word tokens as multiple players into a cooperative game and quantify game-theoretic interactions (*i.e., Shapley Interaction* [12, 40]) among them to investigate the semantic alignment information. LOUPE learns fine-grained semantic alignment from two stages: *token-level Shapley interaction modeling* and *semantics-level Shapley interaction modeling*, where we first learn to identify semantic regions of images that correspond to some semantically meaningful entities, and then align these regions with phrases in the paired text.

Specifically, *token-level Shapley interaction modeling* aims to group patch tokens of images into semantic regions that semantically correspond to some visual instances. From the game-theoretic view, we take patch tokens as players and the similarity score between images and texts as the game function. Intuitively, supposing a set of patch tokens correspond to a visual instance in the image, then they tend to have strong interaction to form the complete semantics of the corresponding instance, which contributes to the better similarity judgment with the paired text. Based on this insight, we take the token-level Shapley interaction as soft supervision labels to encourage the model to capture semantic regions from images. Then, *semantics-level Shapley interaction modeling* infers the fine-grained semantic alignment between semantic regions and phrases. We consider every region and phrase as players and define a fine-grained similarity score as the game function. If a region and a phrase have strong correspondence, they tend to interact with each other and contribute to the fine-grained similarity score. By measuring the Shapley interaction between each region-phrase pair, we obtain the alignment information to guide the pre-training model.

As computing the exact Shapley interaction is an NP-hard problem [32], existing methods mainly employ sampling-based method [6] to obtain unbiased estimation. However, as the number of players grows, they require thousands of model evaluations. To reduce the computational cost, we further propose an efficient hybrid Shapley interaction learning strategy, where an uncertainty-aware neural Shapley interaction learning module cooperates with the sampling-based method. Experimental results show that our hybrid strategy significantly saves the computational cost while maintaining the estimation accuracy. More analysis is shown in Section 4.5.

Our framework serves as a proxy training objective that explicitly establishes the fine-grained semantic alignment between local region and phrase representations. This proxy objective can be directly removed for downstream tasks, rendering an efficient and semantics-sensitive *dual-encoder* model. Experiments show that LOUPE achieves new state-of-the-art on image-text retrieval benchmarks. For text-to-image retrieval on MSCOCO, LOUPE surpasses its strongest competitor by 4.2% on recall@1. Further, without any fine-tuning, LOUPE successfully transfers to object detection and visual grounding tasks in a zero-shot manner. For object detection, it achieves 12.1% mAP on COCO and 19.5% mAP on PASCAL VOC. For visual grounding, it achieves 26.8% accuracy on RefCOCO and 23.6% accuracy on RefCOCO+. Our contributions are summarized as follows:

- We propose **LOUPE**🔍 that explicitly learns fine-grained semantic alignment between visual regions and textual phrases while preserving the high retrieval efficiency of dual-encoder.

- We introduce an efficient and effective hybrid Shapley interaction learning strategy, based on an uncertainty-aware neural Shapley interaction learning module and a sampling-based method.

- Pre-trained on image-text data, LOUPE achieves new state-of-the-art on image-text retrieval and successfully transfers to the tasks that require more fine-grained object-level visual understanding (*i.e., object detection and visual grounding*) without any fine-tuning.

- As manual annotations for masses of object categories is time-consuming and unscalable, our work demonstrates a promising alternative, that is, learning fine-grained semantics from raw texts about images, which are easily available and contain a broader set of visual concepts.

## 2 Related Work

**Vision-Language Pre-Training.** The great success of pre-train-and-fine-tune paradigm in natural language processing [5, 9] and computer vision [10, 14, 47] has been expanded to the joint domain of vision and language [2, 3, 22]. Dominant vision-language pre-training models can be categorized into two groups: *dual-encoder* and *fusion-encoder*. The dual-encoder methods [17, 26, 35, 49] adopt two individual encoders to embed images and texts separately, and model the cross-modal interaction by cosine similarity. Such architecture is efficient for large-scale image-text retrieval as image and text representations can be pre-computed offline. However, simply measuring the cosine similarity between global representations is shallow to capture fine-grained semantic relationships between regions and phrases. The fusion-encoder methods [7, 15, 16, 19, 20, 24, 25, 30, 34, 42, 43, 50, 55] adopt a single multi-modal encoder to jointly model the concatenated sequence of images and texts, which achieves deeper cross-modality interaction. However, these methods are less efficient as images and texts are intertwined to compute the cross-modal attention and can not be pre-computed offline. Further, there are no explicit supervision signals to encourage the alignment between regions and phrases. Some works [7, 24, 25, 30, 43, 50, 55, 56] attempt to leverage an off-the-shelf object detector to extract object features for pre-training. However, the detector is usually pre-trained on limited object categories. Furthermore, considering the excessive demand on memory and computation, existing methods usually fix the parameters of detection models and regard region detection as a pre-processing step, disconnected with vision-language pre-training. Thus, the performance is also restricted by the quality of detection models. FILIP [49] uses a token-wise maximum similarity to enhance the cross-modal interaction of dual-encoder methods. To learn explicit fine-grained semantic alignment, GLIP [23] and X-VLM [53] utilize human-annotated datasets, where regions with bounding-box annotations are aligned with text descriptions. Such a manner is time-consuming and hard to scale to larger raw image-text data from the Internet. In contrast, our proposed framework explicitly learns the fine-grained semantic alignment from raw image-text data and at the same time maintains the high efficiency of dual-encoder. Detailed discussions can be found in Appendix K.

**Shapley Values.** The Shapley value [40] was initially introduced in game theory. It has been theoretically proven to be the unique metric to fairly estimate the contribution of each player in a cooperative game such that certain desirable axioms are satisfied [46]. With solid theoretic foundations, Shapley value has recently been studied as post-hoc explanation methods for Deep Neural Networks (DNN) [8, 31, 54]. Lundberg *et al.* [31] propose a unified attribution method based on Shapley value to interpret the predictions of DNN. Ren *et al.* [38] propose to explain adversarial attacks by Shapley value. In this paper, we propose to model fine-grained semantic alignment by game-theoretic interactions, along with an efficient Shapley interaction learning strategy.

## 3 Method

In this section, we first introduce the problem formulation of fine-grained semantically aligned vision-language pre-training in Section 3.1. Then, we propose the corresponding LOUPE framework for fine-grained semantic alignment learning in Section 3.2 and an efficient approach for Shapley interaction learning in Section 3.3.

### 3.1 Problem Formulation and Model Overview

Generally, vision-language pre-training aims to learn an image encoder $f_\mathrm{I}$ and a text encoder $f_\mathrm{T}$ by cross-modal contrastive learning, where the matched image-text pairs are optimized to get closer and the mismatched pairs are optimized to get further. Let $f_\mathrm{I}(I_i)$ and $f_\mathrm{T}(T_i)$ denote the global representations of the image and text. Then the cross-modal contrastive loss can be formulated as:

$$\mathcal{L}_\mathrm{CMC} = -\log \frac{\exp(f_\mathrm{I}(I_i)^\top f_\mathrm{T}(T_i)/\tau))}{\sum_j^B \exp(f_\mathrm{I}(I_i)^\top f_\mathrm{T}(T_j)/\tau)} - \log \frac{\exp(f_\mathrm{I}(I_i)^\top f_\mathrm{T}(T_i)/\tau))}{\sum_j^B \exp(f_\mathrm{I}(I_j)^\top f_\mathrm{T}(T_i)/\tau))} \tag{1}$$

where $B$ is the batch size and $\tau$ is the temperature hyper-parameter.

While intuitive, such a manner can only learn coarse alignment between images and texts but fails to explicitly capture the fine-grained semantic alignment between visual regions and textual phrases. To learn fine-grained semantic alignment while simultaneously maintaining high retrieval efficiency, we

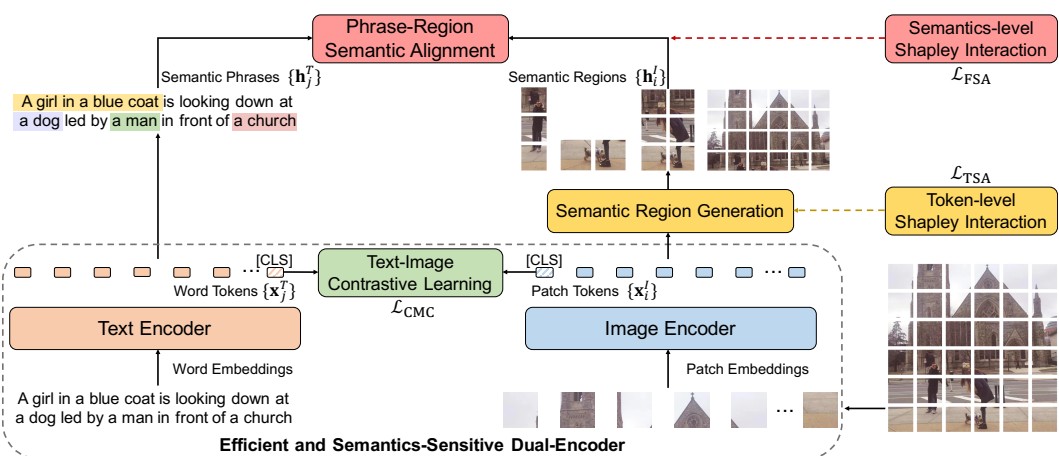

Figure 1: Overview of LOUPE. Our framework serves as a proxy training objective that encourages the image encoder to capture semantic regions and establishes the semantic alignment between region and phrase representations. The proxy training objective can be easily removed for downstream tasks, rendering an efficient and semantics-sensitive dual-encoder.

propose LOUPE, a fine-grained semantically aligned vision-language pre-training framework that germinates from cooperative game theory.

As illustrated in Figure 1, LOUPE learns fine-grained semantic alignment from two stages: **token-level Shapley interaction modeling** and **semantics-level Shapley interaction modeling**. For token-level Shapley interaction modeling, we learn to aggregate patch tokens of images into semantic regions that semantically correspond to some visual concepts, under the guidance of token-based semantic aggregation loss $\mathcal{L}_{\text{TSA}}$. As for semantics-level Shapley interaction modeling, the semantic alignment between the aggregated regions and textual phrases is learned, supervised by the fine-grained semantic alignment loss $\mathcal{L}_{\text{FSA}}$. Combined with the two newly proposed losses, the full objective of fine-grained semantically aligned vision-language pre-training can be formulated as:

$$\mathcal{L} = \mathcal{L}_{\text{CMC}} + \mathcal{L}_{\text{TSA}} + \mathcal{L}_{\text{FSA}} \tag{2}$$

Such a new pre-training objective enforces the image encoder to capture semantic regions and establishes fine-grained semantic alignment between visual regions and textual phrases. During inference, it can be directly removed, rendering an efficient and semantics-sensitive *dual-encoder*.

### 3.2 Interpreting Fine-Grained Semantic Alignment as Game-Theoretic Interaction

#### 3.2.1 Preliminaries

**Shapley Values.** The Shapley value [40] is a classic game theory solution for the unbiased estimation of the importance or contribution of each player in a cooperative game. Considering a game with $\mathcal{N} = \{1, ..., n\}$ players, $\mathcal{S} \subseteq \mathcal{N}$ denotes a potential subset of players. A game $v(\cdot)$ is implemented as a function that maps each subset $\mathcal{S}$ of players to a score, modeling the outcome of a game when players in $\mathcal{S}$ participate in. Specifically, $v(\mathcal{N}) - v(\varnothing)$ denotes the contribution obtained by all players in the game. The Shapley value $\phi(i|\mathcal{N})$ for player $i$ is defined as the average marginal contribution of player $i$ to all possible coalitions $\mathcal{S}$ that are formed without $i$:

$$\phi(i|\mathcal{N}) = \sum_{\mathcal{S} \subseteq \mathcal{N} \setminus \{i\}} p(\mathcal{S})[v(\mathcal{S} \cup \{i\}) - v(\mathcal{S})], \quad p(\mathcal{S}) = \frac{|\mathcal{S}|!(|\mathcal{N}| - |\mathcal{S}| - 1)!}{|\mathcal{N}|!} \tag{3}$$

where $p(\mathcal{S})$ is the likelihood of $\mathcal{S}$ being sampled. The Shapley value has been proved to be the unique metric that satisfies the following axioms: *Linearity*, *Symmetry*, *Dummy*, and *Efficiency* [46]. We summarize these axioms in Appendix B.

**Shapley Interaction.** In the game theory, some players tend to form a coalition and always participate in the game together. The players in the coalition might interact or cooperate with each other, which

brings additional contributions to the game. The Shapley interaction [12] measures this additional contributions brought by the coalition compared with the case when the players work individually. For a coalition $\mathcal{S}$, we consider $[\mathcal{S}]$ as a single hypothetical player, which is the union of the players in $\mathcal{S}$. Then, the reduced game is formed by removing the individual players in $\mathcal{S}$ from the game and adding $[\mathcal{S}]$ to the game. The Shapley value $\phi([\mathcal{S}]|\mathcal{N} \setminus \mathcal{S} \cup \{[\mathcal{S}]\})$ for player $[\mathcal{S}]$ can be computed using Equation 3 over the reduced game. Similarly, we can obtain $\phi(i|\mathcal{N} \setminus \mathcal{S} \cup \{i\})$, where $i$ is the individual player in $S$. Finally, the Shapley interaction for coalition $\mathcal{S}$ is formulated as:

$$\Im([\mathcal{S}]) = \phi([\mathcal{S}]|\mathcal{N} \setminus \mathcal{S} \cup \{[\mathcal{S}]\}) - \sum_{i \in \mathcal{S}} \phi(i|\mathcal{N} \setminus \mathcal{S} \cup \{i\}) \tag{4}$$

In this way, $\Im([\mathcal{S}])$ reflects the interactions inside $\mathcal{S}$. The higher value of $\Im([\mathcal{S}])$ indicates that players in $\mathcal{S}$ cooperate closely with each other.

### 3.2.2 Token-Level Shapley Interaction Modeling

Due to inherent semantic unit mismatch between texts and images, it is ineffective to directly compute the alignment between words and pixels (patches). A textual phrase usually refers to a specific image region, which is composed of multiple patches and represents a visual instance. Thus, we first introduce token-level Shapley interaction modeling to aggregate patches into semantic regions.

**Input Representations.** Given an image-text pair, the input image $I$ is sliced into patches and flattened. Followed by linear projection layer and position embeddings, we obtain patch token sequence $\mathcal{X}^I = \{\mathbf{x}_i^I\}_{i=1}^{L_1}$ with an additional [CLS_I] token embedding. The input text $T$ is tokenized and embedded into word token sequence $\mathcal{X}^T = \{\mathbf{x}_i^T\}_{i=1}^{L_2}$, added with position embeddings. We also prepend a learnable special token [CLS_T] to the word token sequence. Then, we adopt a dual-encoder structure to encode the patch token sequence and word token sequence, separately. On top of the image and text encoders, we obtain the representations of patch token sequence $\tilde{\mathcal{X}}^I = \{\tilde{\mathbf{x}}_i^I\}_{i=1}^{\tilde{L}_1}$ and word token sequence $\tilde{\mathcal{X}}^T = \{\tilde{\mathbf{x}}_i^T\}_{i=1}^{\tilde{L}_2}$. We take the learned representations of [CLS_I] and [CLS_T] tokens as the global representations for images and texts. And the global similarity of image-text pairs is measured by the cosine similarity between them.

**Understanding Semantic Region via Shapley Interaction.** Supposing a set of patches represent a complete visual instance in an image, then they tend to have a strong Shapley interaction because they work jointly to form a visual instance, which contributes to the better similarity judgment with the text. From the game-theoretic view, we take patch tokens and word tokens as players $\mathcal{X} = \mathcal{X}^I \cup \mathcal{X}^T$, and the global similarity between images and texts as the game score $v_1(\cdot)$. To compute $v_1(\mathcal{S})$, we keep tokens in $\mathcal{S}$ and mask input tokens in $\mathcal{X} \setminus \mathcal{S}$ to zeros. Thus, the global similarity only considers the tokens in $\mathcal{S}$, which reflects the contribution of the tokens in $\mathcal{S}$ to the global similarity judgment.

**Semantic Region Generation.** Inspired by *YOLOv3* [37], we design a lightweight region generation module. It takes each patch token representation $\tilde{\mathbf{x}}_i^I$ as input and generates a bounding box prediction centered on $\tilde{\mathbf{x}}_i^I$, which corresponds to a visual region $\mathcal{R}_i = \{\mathbf{x}_{i,k}^I\}_{k=1}^{K_i}$ with $K_i$ patch tokens. The region generation module also predicts a confidence score $s(\mathcal{R}_i)$ for each region. We select the top-$M$ predictions as the semantic regions. Then, the Shapley interaction of $\mathcal{R}_i$ can be defined as:

$$\Im([\mathcal{R}_i]) = \phi([\mathcal{R}_i]|X \setminus \mathcal{R}_i \cup \{[\mathcal{R}_i]\}) - \sum_{\mathbf{x}_{i,k}^I \in \mathcal{R}_i} \phi(\mathbf{x}_{i,k}^I|\mathcal{X} \setminus \mathcal{R}_i \cup \{\mathbf{x}_{i,k}^I\}) \tag{5}$$

According to the Equation 3, we can reformulate Shapley value into the form of expectation:

$$\phi([\mathcal{R}_i]|\mathcal{X} \setminus \mathcal{R}_i \cup \{[\mathcal{R}_i]\}) = \mathbb{E}\{\underset{\substack{\mathcal{S} \subseteq \mathcal{X}^T \setminus \mathcal{R}_i \\ |\mathcal{S}|=c}}{\mathbb{E}} [v_1(\mathcal{S} \cup \mathcal{R}_i) - v_1(\mathcal{S})]\} \tag{6}$$

where $c$ represents the coalition size. $\phi(\mathbf{x}_{i,k}^I|X \setminus \mathcal{R}_i \cup \{\mathbf{x}_{i,k}^I\})$ can be defined in a similar manner, and the Shapley interaction of $\mathcal{R}_i$ can be reformulated as (we provide the proof in Appendix C):

$$\Im([\mathcal{R}_i]) = \mathbb{E}\{\underset{\substack{\mathcal{S} \subseteq \mathcal{X} \setminus \mathcal{R}_i \\ |\mathcal{S}|=c}}{\mathbb{E}} [v_1(\mathcal{S} \cup \mathcal{R}_i) - \sum_{\mathbf{x}_{i,k}^I \in \mathcal{R}_i} v_1(\mathcal{S} \cup \{\mathbf{x}_{i,k}^I\}) + (K-1)v_1(\mathcal{S})]\} \tag{7}$$

Taking normalized $\mathfrak{I}'([R_i])$ as the soft supervision label, the token-based semantic aggregation loss is defined as cross-entropy loss:

$$\mathcal{L}_{\text{TSA}} = -\frac{1}{M}\sum_{i=1}^{M}[\mathfrak{I}'([\mathcal{R}_i])\log(s(\mathcal{R}_i)) + (1 - \mathfrak{I}'([\mathcal{R}_i]))\log(1 - s(\mathcal{R}_i))] \tag{8}$$

which propagates gradients to the region generation module and image encoder to adjust bounding box predictions such that more accurate semantic regions can be captured.

### 3.2.3 Semantics-Level Shapley Interaction Modeling

After obtaining the inferred semantic regions, we propose semantics-level Shapley interaction modeling to explicitly model the fine-grained semantic alignment between regions and phrases. We first define the fine-grained similarity score and then explain semantic alignment based on game theory.

We adopt `Avg-Pooling` over learned patch representations in each $\mathcal{R}_i$ to obtain region representation $\mathbf{h}_i^I \in \mathbb{R}^d$. We employ an off-the-shelf constituency parser to extract phrases from text and obtain phrase representation $\mathbf{h}_i^T \in \mathbb{R}^d$ by `Avg-Pooling`. Totally, we obtain $M$ regions $\mathcal{H}^I = \{\mathbf{h}_i^I\}_{i=1}^M$ and $N$ phrases $\mathcal{H}^T = \{\mathbf{h}_j^T\}_{j=1}^N$. And the alignment matrix can be defined as: $\mathbf{A} = [a_{ij}]^{M \times N}$, where $a_{ij} = \mathbf{h}_i^{I\top}\mathbf{h}_j^T$ represents the alignment score between $i$-th region and $j$-th phrase. Next, we apply softmax-normalization over each row of $\mathbf{A}$, obtaining $\tilde{\mathbf{A}}$. For the $i$-th region, we calculate its maximum alignment score as $\max_j \tilde{a}_{ij}$. Then, we use the average maximum alignment score over all regions as the fine-grained image-to-text similarity $p_1$. Similarly, we can obtain the fine-grained text-to-image similarity $p_2$, and the total fine-grained similarity score can be defined: $p = (p_1 + p_2)/2$.

**Understanding Semantic Alignment via Shapley Interaction.** If a region and a phrase have strong semantic correspondence, then they tend to cooperate with each other and contribute to the fine-grained similarity score. Thus, we can consider $\mathcal{H} = \mathcal{H}^I \cup \mathcal{H}^T$ as the players and the fine-grained similarity score $p$ as the game score $v_2(\cdot)$. The Shapley interaction of them can be formulated as:

$$\mathfrak{I}([\mathcal{H}_{ij}]) = \phi([\mathcal{H}_{ij}]|\mathcal{H} \setminus \mathcal{H}_{ij} \cup \{[\mathcal{H}_{ij}]\}) - \phi(\mathbf{h}_i^I|\mathcal{H} \setminus \mathcal{H}_{ij} \cup \{\mathbf{h}_i^I\}) - \phi(\mathbf{h}_j^T|\mathcal{H} \setminus \mathcal{H}_{ij} \cup \{\mathbf{h}_j^T\}) \tag{9}$$

$$= \mathbb{E}_c\{\underset{\substack{\mathcal{S} \subseteq \mathcal{H} \setminus \mathcal{H}_{ij} \\ |\mathcal{S}| = c}}{\mathbb{E}}[v_2(\mathcal{S} \cup \mathcal{H}_{ij}) - v_2(\mathcal{S} \cup \{\mathbf{h}_i^I\}) - v_2(\mathcal{S} \cup \{\mathbf{h}_j^T\}) + v_2(\mathcal{S})]\} \tag{10}$$

where $[\mathcal{H}_{ij}]$ represents the single player formed by the coalition of $i$-th region and $j$-th phrase. Taking normalized $\mathfrak{I}'([\mathcal{H}_{ij}])$ as soft labels, the fine-grained semantic alignment loss can be defined as:

$$\mathcal{L}_{\text{FSA}} = -\frac{1}{MN}\sum_{i=1}^{M}\sum_{j=1}^{N}\mathfrak{I}'([\mathcal{H}_{ij}])\log(\tilde{a}_{ij}) \tag{11}$$

### 3.3 Uncertainty-Aware Neural Shapley Interaction Learning

According to Equation 3 and Equation 4, computing exact the Shapley value is an NP-hard problem [32]. Previous methods mainly apply sampling-based method [6] to approximate it. While sampling-based approximation is unbiased, an accurate approximation requires thousands of model evaluations. To reduce the computational cost, we propose an uncertainty-aware neural Shapley interaction learning (UNSIL) module to cooperate with the sampling-based method, rendering an efficient and effective hybrid strategy.

Specifically, the sampling-based method [6] estimates the expectation terms in Equation 7 and Equation 10 by sampling to compute the Shapley interaction. Inspired by noisy label learning [18], the UNSIL module learns to predict the Shapley interaction and the corresponding uncertainty $\sigma \in (0, 1)$. Intuitively, if the UNSIL module makes a prediction with low uncertainty, we can directly apply its prediction to $\mathcal{L}_{\text{TSA}}$ and $\mathcal{L}_{\text{FSA}}$, avoiding thousands of model evaluations. If the uncertainty is high, we then resort to the sampling-based method for a more accurate estimation.

During training, the UNSIL module first predicts the target interaction with uncertainty $\sigma$. Then, we sample a value $\epsilon$ from a uniform distribution on $(0, 1)$. If $\epsilon > \sigma$, we directly use its prediction. If

Table 1: Results (%) of zero-shot image-text retrieval on Flickr30K and MSCOCO datasets.

| | Flickr30K | | | | | | MSCOCO | | | | | |
| | image-to-text | | | text-to-image | | | image-to-text | | | text-to-image | | |
| | R@1 | R@5 | R@10 | R@1 | R@5 | R@10 | R@1 | R@5 | R@10 | R@1 | R@5 | R@10 |
|---|---|---|---|---|---|---|---|---|---|---|---|---|
| ImageBERT | 70.7 | 90.2 | 94.0 | 54.3 | 79.6 | 87.5 | 44.0 | 71.2 | 80.4 | 32.3 | 59.0 | 70.2 |
| UNITER | 83.6 | 95.7 | 97.7 | 68.7 | 89.2 | 93.9 | - | - | - | - | - | - |
| CLIP | 88.0 | 98.7 | 99.4 | 68.7 | 90.6 | 95.2 | 58.4 | 81.5 | 88.1 | 37.8 | 62.4 | 72.2 |
| ALIGN | 88.6 | 98.7 | 99.7 | 75.7 | 93.8 | **96.8** | 58.6 | 83.0 | 89.7 | 45.6 | 69.8 | 78.6 |
| FILIP | 89.8 | 99.2 | 99.8 | 75.0 | 93.4 | 96.3 | 61.3 | 84.3 | 90.4 | 45.9 | 70.6 | 79.3 |
| **LOUPE** | **90.5** | **99.5** | **99.8** | **76.3** | **93.9** | 96.7 | **62.3** | **85.1** | **91.2** | **50.1** | **75.4** | **83.3** |

Table 2: Top-1 accuracy (%) of zero-shot image classification over 11 datasets.

| | CIFAR10 | Food101 | StanfordCars | SUN397 | Flowers102 | Country211 | FER2013 | Aircrafts | OxfordPets | Caltech101 | ImageNet | Average |
|---|---|---|---|---|---|---|---|---|---|---|---|---|
| CLIP | **96.2** | 92.9 | 77.3 | 67.7 | 78.7 | 34.9 | **57.7** | 36.1 | 93.5 | 92.6 | 75.3 | 73.0 |
| **LOUPE** | 95.9 | **94.3** | **79.9** | **69.8** | **87.4** | **37.8** | 53.3 | **54.9** | **94.1** | **93.9** | **76.1** | **76.1** |

$\epsilon \leq \sigma$, we then use the sampling-based method to compute the Shapley Interaction and update the UNSIL module based on the sampling-based results. Note that, for the first few iterations, we employ the sampling-based method directly, and use its results to train the UNSIL module.

Let $\mathfrak{I}^*$ and $\hat{\mathfrak{I}}$ denote the results from the sampling-based method and UNSIL module, respectively. Taking $\mathfrak{I}^*$ as the ground-truth, the UNSIL module is trained by:

$$\mathcal{L}_{\text{UNSIL}} = \frac{1}{\beta_1 \sigma} \mathcal{L}_{\text{MSE}}(\hat{\mathfrak{I}}, \mathfrak{I}^*) + \beta_2 \sigma \tag{12}$$

where the first term is mean squared error $\mathcal{L}_{\text{MSE}}$ weighted by the uncertainty, the second term serves as a regularization term for the prediction uncertainty, and $\beta$ is the weight hyper-parameter. The UNSIL module implicitly learns the uncertainty from the regression loss function. We discuss the implementation details of the UNSIL module in Section 4.5 and Appendix D.

## 4 Experiments

### 4.1 Pre-training Details

As sufficient data is a prerequisite for vision-language pre-training, we construct a dataset with 240M image-text pairs from the Internet. We implement the image encoder by Swin-L [28] and the text encoder by BERT-Small [9]. The input images are resized to $224 \times 224$ and the input texts are tokenized by WordPiece with a maximum length of 60. We pre-train the model for 20 epochs using a batch size of 512 on 128 NVIDIA V100 GPUs. We utilize AdamW [29] optimizer with a learning rate of $2 \times 10^{-4}$ and a weight decay of 0.01. More pre-training and evaluation details are provided in Appendix D, E. We also analyze the image encoder and training efficiency in Appendix G, J.

### 4.2 Zero-Shot Image-Text Retrieval

We compare LOUPE on the widely used MSCOCO [27] and Flickr30K [33] datasets. First, the results in Table 1 show that LOUPE achieves new state-of-the-art zero-shot performance on most metrics of the two datasets, demonstrating the stronger generalizability of our pre-training framework. Second, while previous works mainly pre-train on larger datasets (CLIP 400M, ALIGN 1800M, FILIP 340M), LOUPE still achieves superior performance using less training data (240M). Third, compared with FILIP which directly computes token-wise similarity, our model captures semantic alignment between visual regions and textual phrases, which is more semantically meaningful. For text-to-image retrieval on MSCOCO, LOUPE significantly outperforms FILIP by 4.2% on recall@1.

### 4.3 Zero-Shot Image Classification

In this section, we evaluate LOUPE on the zero-shot image classification task. We compare LOUPE with CLIP on 11 downstream classification datasets, following the same evaluation setting as

Table 3: Without fine-tuning, zero-shot transfer performance on object detection and visual grounding.

| | COCO | | PASCAL VOC | | RefCOCO | | | RefCOCO+ | | |
|---|---|---|---|---|---|---|---|---|---|---|
| | mAP@0.3 | mAP@0.5 | mAP@0.3 | mAP@0.5 | val | testA | testB | val | testA | testB |
| CLIP + Pixel-Wise | 8.5 | 4.5 | 18.2 | 7.3 | 6.7 | 6.2 | 5.8 | 6.1 | 7.0 | 5.7 |
| CLIP + K-Means | 6.4 | 1.9 | 11.7 | 4.8 | 2.1 | 2.3 | 1.7 | 1.7 | 2.0 | 2.8 |
| CLIP + Grad-CAM | 7.1 | 3.2 | 19.1 | 8.2 | 5.5 | 5.2 | 4.8 | 4.4 | 5.6 | 4.9 |
| AdaptCLIP | 14.9 | 6.6 | 28.7 | 12.9 | 16.7 | 18.4 | 18.0 | 17.5 | 18.9 | 19.6 |
| **LOUPE** | **25.3** | **12.1** | **30.3** | **19.5** | **25.2** | **26.8** | **24.5** | **22.9** | **23.3** | **23.6** |

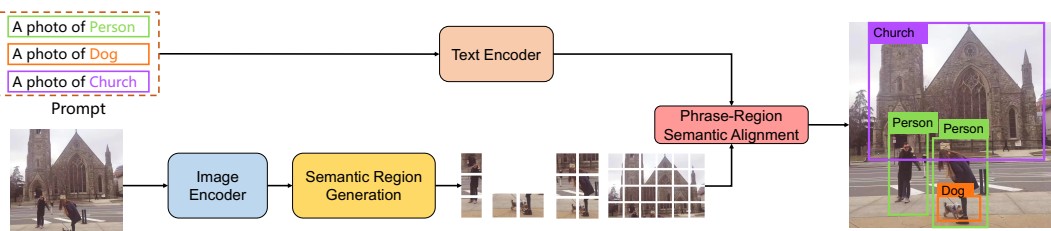

Figure 2: An example of LOUPE zero-shot transferring to object detection using prompt templates.

CLIP [35]. Table 2 summarizes the results. As shown in Table 2, our LOUPE outperforms CLIP with average improvement of 3.1%. Notably, on ImageNet, the largest dataset among 11 datasets, our LOUPE surpasses CLIP by 0.8%. Also, we observe that LOUPE achieves substantial performance gains on several fine-grained image classification datasets (i.e., Flowers102 and Aircrafts). It demonstrates the superiority of our LOUPE on fine-grained semantics understanding.

We also evaluate the linear probing performance of our LOUPE on image classification. The detailed results can be found in Appendix I.

## 4.4 Zero-Shot Transfer to Object Detection and Visual Grounding

To answer whether our model has learned fine-grained semantics, we further evaluate LOUPE on object detection [39] and visual grounding [51], which require more fine-grained semantic understanding ability to identify specific visual regions in images according to the object labels or referring expressions. Visual grounding can be seen as generalized object detection, where the pre-defined class labels are replaced by language referring expression sentences. As LOUPE can generate a set of semantic regions that are aligned with textual phrases, it can be easily applied to object detection and visual grounding without structure modification. For visual grounding, we take referring expressions as input text. For object detection, as illustrated in Figure 2, we use prompt to expand detection labels to input text. Then, we encode input text by the learned text encoder, and these tasks can be completed by measuring the similarity between candidate region representations and text representations.

For comparison, we zero-shot transfer CLIP (ViT-L/14) to object detection and visual grounding by applying several non-parametric approaches on the spatial feature maps of CLIP. We also compare with AdaptCLIP [21], which is a concurrently unpublished method that leverages classic super-pixel (SLIC [1]) and bounding box proposal (selective search [44]) methods to zero-shot transfer CLIP to phrase localization. We use its public official implementations to get the experiment results. For object detection, we evaluate their mean Average Precision (mAP) at IoU thresholds of $\{0.3, 0.5\}$ on COCO [27] (65 classes) and PASCAL VOC [11] (20 classes). For visual grounding, we evaluate their top-1 accuracy at IoU thresholds of 0.5 on RefCOCO [51] and RefCOCO+ [51]. The experiment details of CLIP variants and LOUPE are provided in Appendix E.

Table 3 summarizes the results. **1)** Overall, LOUPE outperforms all CLIP variants by a large margin. The significantly higher performance illustrates the stronger zero-shot transfer ability of our fine-grained semantically aligned pre-training paradigm. **2)** Second, all CLIP variants rely on pre-processing steps on CLIP's feature map (*e.g.*, AdaptCLIP first uses SLIC to group pixels and then uses selective search to generate a large number of proposals), which is time-consuming. In contrast, our method directly predicts the semantic regions based on the patch token representations. **3)** Third, the consistently competitive performance across four benchmarks validates that LOUPE can learn fine-grained semantics from raw text supervision. LOUPE demonstrates a promising alternative, that is, learning fine-grained semantics from large-scale raw image-text pairs, which are easily available and contain a broader set of visual concepts.

Table 4: Ablation study of each component across three tasks.

| | | MSCOCO | | COCO | | RefCOCO | | | Training Time |
|---|---|---|---|---|---|---|---|---|---|
| | | I2T | T2I | mAP@0.3 | mAP@0.5 | val | testA | testB | (sec/iter) |
| 1 | Backbone | 31.0 | 24.8 | 3.8 | 1.0 | 1.3 | 0.9 | 0.8 | 1.17 |
| 2 | Backbone + $\mathcal{L}_{\text{TSA}}$ | 32.4 | 26.2 | 7.6 | 3.3 | 1.8 | 2.0 | 2.6 | 8.38 |
| 3 | Backbone + $\mathcal{L}_{\text{TSA}}$ + $\mathcal{L}_{\text{FSA}}$ | 33.5 | 28.3 | 9.4 | 5.9 | 4.1 | 4.6 | 4.3 | 9.90 |
| 4 | Backbone + $\mathcal{L}_{\text{TSA}}$ + $\mathcal{L}_{\text{FSA}}$ + UNSIL | 33.3 | 28.1 | 9.0 | 5.6 | 4.5 | 4.9 | 4.4 | 1.93 |

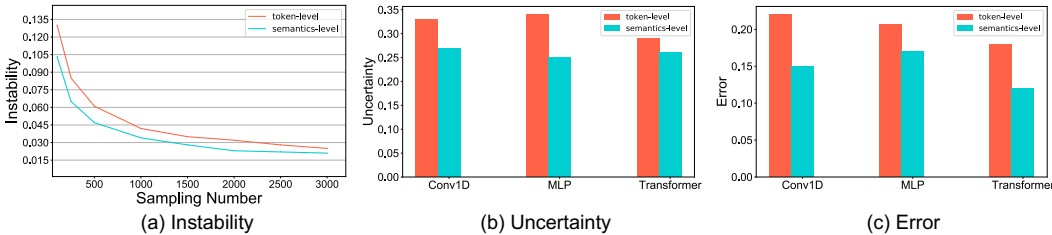

(a) Instability  (b) Uncertainty  (c) Error

Figure 3: (a) Instability of the Shapley interaction approximation with respect to different sampling numbers. (b, c) Uncertainty and error of the UNSIL module with different structures.

As time-consuming human annotations are unscalable for massive object classes in the real world, some recent works [4, 36] target at training object detectors with annotations on base object classes to generalize to the remaining object classes of the same dataset. The latest works [13, 52] leverage the generalizability of vision-language pre-training models to further improve the zero-shot performance on novel classes. However, these zero-shot approaches still require bounding box annotations on base classes for task-specific supervised learning. In contrast, our LOUPE is trained on large-scale raw image-text pairs, which are already accessible on the Internet and contain more diverse semantics.

## 4.5 Ablation Study

**Effectiveness of Individual Components.** In this section, we investigate the effectiveness of each component in Table 4. Given the costly training time, all ablation studies are based on a relatively small dataset (Conceptual Captions 3M [41]). We start with the backbone model that consists of a dual-encoder trained by cross-modal contrastive loss. We then gradually add token-level Shapley interaction modeling supervision $\mathcal{L}_{\text{TSA}}$ (Row 2), semantics-level Shapley interaction modeling supervision $\mathcal{L}_{\text{FSA}}$ (Row 3), and UNSIL module (Row 4). For Row 2 and Row 3, the Shapley interaction is only computed by the sampling-based method. The results in Table 4 show that both $\mathcal{L}_{\text{TSA}}$ and $\mathcal{L}_{\text{FSA}}$ bring significant improvement for all tasks. We observe that $\mathcal{L}_{\text{TSA}}$ boosts a 3.8% improvement on object detection. And the improved fine-grained visual semantic understanding further facilitates the cross-modal retrieval performance (+1.4%). The semantics-level Shapley interaction modeling further improves the performance on all tasks by modeling the semantic alignment between visual regions and textual phrases. Comparing Row 3 and Row 4, we observe that the UNSIL module maintains the estimation accuracy while avoiding intensive computations. The averaged training time is reduced from 9.90 seconds per iteration to 1.93 seconds per iteration.

**Accuracy of the Shapley Interaction Learning.** Since we use the sampling-based method [6] to compute the Shapley Interaction and train the UNSIL module, we conduct a study to evaluate the accuracy of the sampling-based method and the error of the UNSIL module. As [54], we compute the interaction multiple times and measure the instability of them. A lower instability means that we obtain similar interactions from different sampling processes. It indicates a high accuracy. Specifically, the instability is defined as $\frac{\mathbb{E}_{u,v:u\neq v}|\mathfrak{I}_u-\mathfrak{I}_v|}{\mathbb{E}_w|\mathfrak{I}_w|}$, where $\mathfrak{I}_w$ denotes the interaction computed in the $w$-th time. We average the instability values over Shapley interaction of 100 image-text pairs. We report the average instability values with respect to different sampling numbers. As shown in Figure 3 (a), the instability decreases along with the increase of the sampling number. When the sampling number is larger than 500, the approximated Shapley interaction is stable enough with instability less than 0.06. Further, we attempt different models (*i.e., Conv1D, 3-Layer MLP + Attention, 3-Layer Transformer*) to implement the UNSIL module (please see Appendix D for more details). We test them on 1000 samples and report their mean uncertainty and relative error in Figure 3 (b) and (c). We observe that *MLP + Attention* is good enough to predict the interaction with lower complexity. Thus, we implement the UNSIL module by *MLP + Attention*.

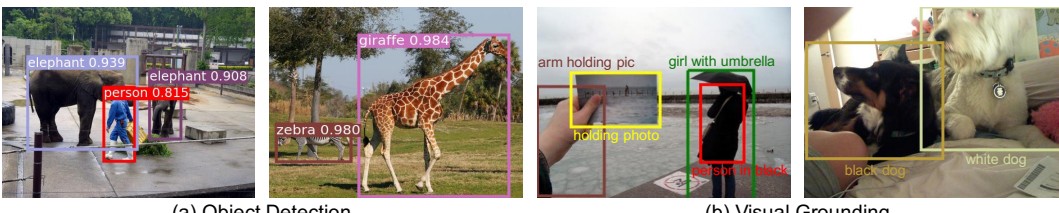

(a) Object Detection      (b) Visual Grounding

Figure 4: Qualitative examples of object detection on COCO and visual grounding on RefCOCO+.

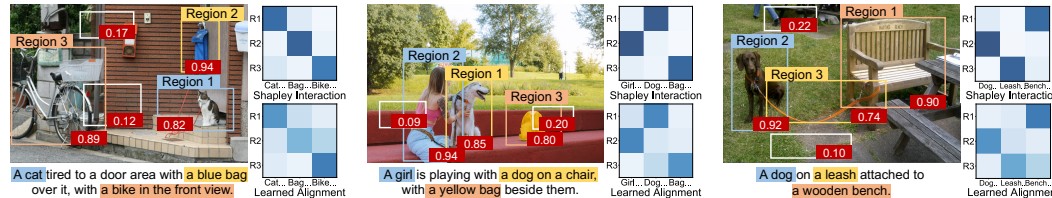

Figure 5: Visualization of learned fine-grained semantic alignment and corresponding Shapley interaction values. The values in the red boxes represent the Shapley interaction of regions.

## 4.6 Qualitative Analysis

**Qualitative Examples.** As shown in Figure 4, LOUPE successfully captures the regions that correspond to the detected objects, and grounds the referring expressions onto the referred regions.

**Visualization of Learned Fine-Grained Semantic Alignment.** In Figure 5, we visualize some key semantic regions and corresponding alignment matrices inferred by LOUPE. We present the regions with top-3 confidence (Region 1 – 3) and two randomly sampled regions (white boxes). The red boxes at the bottom of bounding boxes indicate their normalized token-level Shapley interaction values. Comparing their Shapley interaction values, we observe that the token-level Shapley interaction successfully distinguishes semantic regions from randomly sampled regions. The semantically meaningful regions tend to have stronger interaction. It indicates that token-level Shapley interaction can provide correct supervision for semantic region generation. Further, we show the alignment matrices inferred by semantics-level Shapley interaction and LOUPE, respectively. As shown in the right case of Figure 5, LOUPE successfully recognizes the leash region and aligns it with the "a leash" phrase. Note that existing object detection datasets do not contain the "leash" category.

## 5 Conclusion

This paper introduces a novel vision-language pre-training framework, **LOUPE**, which models the fine-grained semantic alignment between visual regions and textual phrases by game-theoretic interactions. To efficiently compute the interactions, we further propose an uncertainty-aware neural Shapley interaction learning module. Comprehensive experiments show that LOUPE achieves new state-of-the-art on image-text retrieval datasets and can transfer to object detection and visual grounding in a zero-shot manner. This work demonstrates a new promising direction of learning fine-grained semantics from large-scale raw image-text data.

**Limitations.** 1) The phrases are extracted by off-the-shelf constituency parsers, whose predictions might not be completely accurate. 2) The web data might inevitably contain mismatched image-text pairs, leading to noisy supervision.

**Social Impacts.** Our model is trained on noisy data from the Internet, which may contain unsuitable images, violent text, or private information. Thus, additional analysis of the data is necessary. Further, the use of our model for privacy surveillance or other nefarious purposes should be prohibited.

**Acknowledgment.** This work has been supported in part by the National Key Research and Development Program of China (2018AAA0101900), Zhejiang NSF (LR21F020004), Key Research and Development Program of Zhejiang Province, China (No. 2021C01013), Chinese Knowledge Center of Engineering Science and Technology (CKCEST). We thank all the reviewers for their valuable comments.

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
