# A   Appendix Overview

In this appendix, we present:

- Axiomatic Properties of Shapley Value (Section B).

- Proofs of Equation 7 and Equation 10 (Section C).

- Hyperparameters and Implementation Details (Section D).

- Pre-Training and Evaluation Details (Section E).

- More Experiment Results on Downstream Vision-Language Generation Task (Section F).

- Further Analysis on the Image Encoder (Section G).

- More Qualitative Examples on Object Detection and Visual Grounding (Section H).

- Linear Probing Evaluation (Section I).

- Training Efficiency Discussion (Section J).

- Detailed Discussion with Some Related Works (Section K).

# B   Axiomatic Properties of Shapley Value

In this section, we mainly introduce the axiomatic properties of Shapley value. *Weber et al.* [17] have proved that Shapley value is the unique metric that satisfies the following axioms: *Linearity*, *Symmetry*, *Dummy*, and *Efficiency*.

**Linearity Axiom.** If two independent games $u$ and $v$ can be linearly merged into one game $w(\mathcal{S}) = u(\mathcal{S}) + v(\mathcal{S})$, then the Shapley value of each player $i \in \mathcal{N}$ in the new game $w$ is the sum of Shapley values of the player $i$ in the game $u$ and $v$, which can be formulated as:

$$\phi_w(i|\mathcal{N}) = \phi_u(i|\mathcal{N}) + \phi_v(i|\mathcal{N}) \tag{1}$$

**Symmetry Axiom.** Considering two players $i$ and $j$ in a game $v$, if they satisfy:

$$\forall \mathcal{S} \in \mathcal{N} \setminus \{i, j\}, v(\mathcal{S} \cup \{i\}) = v(\mathcal{S} \cup \{j\}) \tag{2}$$

then $\phi_v(i|\mathcal{N}) = \phi_v(j|\mathcal{N})$.

**Dummy Axiom.** The dummy player is defined as the player that has no interaction with other players. Formally, if a player $i$ in a game $v$ satisfies:

$$\forall \mathcal{S} \in \mathcal{N} \setminus \{i\}, v(\mathcal{S} \cup \{i\}) = v(\mathcal{S}) + v(\{i\}) \tag{3}$$

then this player is defined as the dummy player. In this way, the dummy player $i$ has no interaction with other players, *i.e.,* $v(\{i\}) = \phi_v(i|\mathcal{N})$.

**Efficiency Axiom.** The efficiency axiom ensures that the overall reward can be assigned to all players, which can be formulated as:

$$\sum_{i \in \mathcal{N}} \phi_v(i) = v(\mathcal{N}) - v(\varnothing) \tag{4}$$

# C   Proofs of Equation 7 and Equation 10

In this section, we provide detailed proofs for Equation 7 in Section 3.2.2 and Equation 10 in Section 3.2.3.

We first provide proof for Equation 7. The token-level Shapley interaction for $\mathcal{R}_i$ can be decomposed as follows:

$$\mathfrak{I}([\mathcal{R}_i]) = \phi([\mathcal{R}_i]|X \setminus \mathcal{R}_i \cup \{[\mathcal{R}_i]\}) - \sum_{\mathbf{x}_{i,k}^I \in \mathcal{R}_i} \phi(\mathbf{x}_{i,k}^I|\mathcal{X} \setminus \mathcal{R}_i \cup \{\mathbf{x}_{i,k}^I\}) \tag{5}$$

$$= \mathbb{E}_c\{\underset{\substack{\mathcal{S} \subseteq \mathcal{X} \setminus \mathcal{R}_i \\ |\mathcal{S}|=c}}{\mathbb{E}}[v_1(\mathcal{S} \cup \mathcal{R}_i) - v_1(\mathcal{S})]\} - \sum_{\mathbf{x}_{i,k}^I \in \mathcal{R}_i} \mathbb{E}_c\{\underset{\substack{\mathcal{S} \subseteq \mathcal{X} \setminus \mathcal{R}_i \\ |\mathcal{S}|=c}}{\mathbb{E}}[v_1(\mathcal{S} \cup \{\mathbf{x}_{i,k}^I\}) - v_1(\mathcal{S})]\} \tag{6}$$

$$= \mathbb{E}_c\{\underset{\substack{\mathcal{S} \subseteq \mathcal{X} \setminus \mathcal{R}_i \\ |\mathcal{S}|=c}}{\mathbb{E}}[v_1(\mathcal{S} \cup \mathcal{R}_i) - v_1(\mathcal{S})]\} - \mathbb{E}_c\{\underset{\substack{\mathcal{S} \subseteq \mathcal{X} \setminus \mathcal{R}_i \\ |\mathcal{S}|=c}}{\mathbb{E}}[\sum_{\mathbf{x}_{i,k}^I \in \mathcal{R}_i}(v_1(\mathcal{S} \cup \{\mathbf{x}_{i,k}^I\}) - v_1(\mathcal{S}))]\}$$
$$\tag{7}$$

$$= \mathbb{E}_c\{\underset{\substack{\mathcal{S} \subseteq \mathcal{X} \setminus \mathcal{R}_i \\ |\mathcal{S}|=c}}{\mathbb{E}}[v_1(\mathcal{S} \cup \mathcal{R}_i) - v_1(\mathcal{S}) - \sum_{\mathbf{x}_{i,k}^I \in \mathcal{R}_i}(v_1(\mathcal{S} \cup \{\mathbf{x}_{i,k}^I\}) - v_1(\mathcal{S}))]\} \tag{8}$$

$$= \mathbb{E}_c\{\underset{\substack{\mathcal{S} \subseteq \mathcal{X} \setminus \mathcal{R}_i \\ |\mathcal{S}|=c}}{\mathbb{E}}[v_1(\mathcal{S} \cup \mathcal{R}_i) - v_1(\mathcal{S}) - \sum_{\mathbf{x}_{i,k}^I \in \mathcal{R}_i} v_1(\mathcal{S} \cup \{\mathbf{x}_{i,k}^I\}) + \sum_{\mathbf{x}_{i,k}^I \in \mathcal{R}_i} v_1(\mathcal{S})]\} \tag{9}$$

$$= \mathbb{E}_c\{\underset{\substack{\mathcal{S} \subseteq \mathcal{X} \setminus \mathcal{R}_i \\ |\mathcal{S}|=c}}{\mathbb{E}}[v_1(\mathcal{S} \cup \mathcal{R}_i) - \sum_{\mathbf{x}_{i,k}^I \in \mathcal{R}_i} v_1(\mathcal{S} \cup \{\mathbf{x}_{i,k}^I\}) + (K-1)v_1(\mathcal{S})]\} \tag{10}$$

We then provide proof for Equation 10. The semantics-level Shapley interaction between region $i$ and phrase $j$ can be decomposed as follows:

$$\mathfrak{I}([\mathcal{H}_{ij}]) = \phi([\mathcal{H}_{ij}]|\mathcal{H} \setminus \mathcal{H}_{ij} \cup \{[\mathcal{H}_{ij}]\}) - \phi(\mathbf{h}_i^I|\mathcal{H} \setminus \mathcal{H}_{ij} \cup \{\mathbf{h}_i^I\}) - \phi(\mathbf{h}_j^T|\mathcal{H} \setminus \mathcal{H}_{ij} \cup \{\mathbf{h}_j^T\})$$
$$\tag{11}$$

$$= \mathbb{E}_c\{\underset{\substack{\mathcal{S} \subseteq \mathcal{H} \setminus \mathcal{H}_{ij} \\ |\mathcal{S}|=c}}{\mathbb{E}}[v_2(\mathcal{S} \cup \mathcal{H}_{ij}) - v_2(\mathcal{S})]\} - \mathbb{E}_c\{\underset{\substack{\mathcal{S} \subseteq \mathcal{H} \setminus \mathcal{H}_{ij} \\ |\mathcal{S}|=c}}{\mathbb{E}}[v_2(\mathcal{S} \cup \{\mathbf{h}_i^I\}) - v_2(\mathcal{S})]\} \tag{12}$$

$$- \mathbb{E}_c\{\underset{\substack{\mathcal{S} \subseteq \mathcal{H} \setminus \mathcal{H}_{ij} \\ |\mathcal{S}|=c}}{\mathbb{E}}[v_2(\mathcal{S} \cup \{\mathbf{h}_j^T\}) - v_2(\mathcal{S})]\} \tag{13}$$

$$= \mathbb{E}_c\{\underset{\substack{\mathcal{S} \subseteq \mathcal{H} \setminus \mathcal{H}_{ij} \\ |\mathcal{S}|=c}}{\mathbb{E}}[v_2(\mathcal{S} \cup \mathcal{H}_{ij}) - v_2(\mathcal{S} \cup \{\mathbf{h}_i^I\}) - v_2(\mathcal{S} \cup \{\mathbf{h}_j^T\}) + v_2(\mathcal{S})]\} \tag{14}$$

## D   Hyperparameters and Implementation Details

In this section, we summarize the hyperparameters in our LOUPE model in Table 1, including the hyperparameters of the image encoder, text encoder, and pre-training process. For the uncertainty-aware neural Shapley interaction learning module, we attempt three kinds of models (*i.e., Conv1D, 3-Layer MLP + Attention, 3-Layer Transformer*) to implement it for token-level and semantics-level Shapley interaction approximation.

For token-level Shapley interaction approximation, it takes the patch token sequence $\mathcal{X}^I = \{\mathbf{x}_i^I\}_{i=1}^{L_1}$, word token sequence $\mathcal{X}^T = \{\mathbf{x}_i^T\}_{i=1}^{L_2}$, and the visual region $\mathcal{R}_i = \{\mathbf{x}_{i,k}^I\}_{k=1}^{K_i}$ as input, and estimates the corresponding token-level Shapley interaction value for $\mathcal{R}_i$ along with the uncertainty $\sigma$.

**Conv1D** model first performs `Avg-Pooling` over learned patch representations of $\mathcal{R}_i$ to obtain the region representation $\mathbf{h}_i^I$, and then fuse the word and patch token representations with the region representation $\mathbf{h}_i^I$, respectively. Specifically, we project them into an unified semantic space by fully-connected layers and then fuse them through Hadamard product as:

$$\mathcal{F}^I = (\mathcal{W}_1 \mathbf{h}_i^I \mathbf{1}^T) \odot (\mathcal{W}_2 \mathcal{X}^I) \tag{15}$$

where $\mathcal{W}_1$ and $\mathcal{W}_2$ are the learnable projection parameters, $\mathbf{1}^T$ is the transpose of an all-ones vector, and $\odot$ represents Hadamard product. We can obtain $\mathcal{F}^T$ in a similar manner. Then, we apply 1D

Table 1: A summary of various hyperparameters in LOUPE.

| Hyperparameter | Value |
|---|---|
| *Image Encoder - Swin-L* | |
| input image size | $224 \times 224$ |
| stage 1 - patch size | $4 \times 4$ |
| stage 1 - hidden size | 192 |
| stage 1 - window size | $7 \times 7$ |
| stage 1 - number of heads | 6 |
| stage 2 - patch size | $8 \times 8$ |
| stage 2 - hidden size | 384 |
| stage 2 - window size | $7 \times 7$ |
| stage 2 - number of heads | 12 |
| stage 3 - patch size | $16 \times 16$ |
| stage 3 - hidden size | 768 |
| stage 3 - window size | $7 \times 7$ |
| stage 3 - number of heads | 24 |
| stage 4 - patch size | $32 \times 32$ |
| stage 4 - hidden size | 1536 |
| stage 4 - window size | $7 \times 7$ |
| stage 4 - number of heads | 48 |
| *Text Encoder - BERT-Small* | |
| maximum length of word tokens | 60 |
| vocabulary size | 30522 |
| attention dropout probability | 0.1 |
| hidden activation function | GELU |
| hidden dropout probability | 0.1 |
| initializer range | 0.02 |
| intermediate size | 2048 |
| layer norm eps | $1e^{-12}$ |
| hidden size | 512 |
| number of attention heads | 8 |
| number of hidden layers | 4 |
| *Pre-Training* | |
| number of epochs | 20 |
| batch size | 512 |
| learning rate | 2e-4 |
| learning schedule | OneCycle |
| cycle momentum | Ture |
| base momentum | 0.85 |
| max momentum | 0.95 |
| AdamW weight decay | 0.01 |
| AdamW $\beta_1$ | 0.9 |
| AdamW $\beta_2$ | 0.999 |

convolution operation with kernel size = 4 and stride = 2 over $\mathcal{F}^I$ and $\mathcal{F}^T$, respectively. Following with Max-Pooling operation, we obtain $\tilde{\mathbf{f}}^I \in \mathbb{R}^d$ and $\tilde{\mathbf{f}}^T \in \mathbb{R}^d$. Next, we concatenate them with $\mathbf{h}_i^I$ and feed them to two separate 1-layer fully connected layers to get the Shapley interaction estimation and corresponding uncertainty.

**3-Layer MLP + Attention** model first performs Avg-Pooling over learned patch representations of $\mathcal{R}_i$ to obtain the region representation $\mathbf{h}_i^I$. Then, we use $\mathbf{h}_i^I$ as the query to attend the patch token sequence and compute a weighted sum of the patch token representations as:

$$\tilde{\alpha}_j^I = \mathcal{W}_3(tanh(\mathcal{W}_4\mathbf{h}_i^I + \mathcal{W}_5\mathbf{x}_j^I)) \tag{16}$$

$$\alpha^I = softmax([\tilde{\alpha}_1^I, ..., \tilde{\alpha}_{L_1}^I]) \tag{17}$$

$$\mathbf{e}^I = \sum_{j=1} \alpha_i^I \mathbf{x}_j^I \tag{18}$$

Where $L_1$ is the number of patch tokens. We can obtain $\mathbf{e}^T$ for word token sequence in a similar manner. Consequently, we concatenate them and $\mathbf{h}_i^I$ and feed them to two separate 3-layer fully connected layers to get the Shapley interaction estimation and corresponding uncertainty.

**3-Layer Transformer** model takes the concatenated sequence $\mathcal{X}^I$ and $\mathcal{X}^T$ as input. We add position embeddings and three kinds of token type embeddings (*i.e.*, *word token, context patch token, region patch token*) to them. We then apply three layers of transformer blocks to jointly encode the input sequence and take the output `[CLS]` token to predict the Shapley interaction estimation and corresponding uncertainty, separately.

For semantics-level Shapley interaction approximation, it takes the $M$ regions $\mathcal{H}^I = \{\mathbf{h}_i^I\}_{i=1}^M$, $N$ phrases $\mathcal{H}^T = \{\mathbf{h}_j^T\}_{j=1}^N$, and the target region-phrase pair $< \mathbf{h}_i^I, \mathbf{h}_j^T >$ as input, and estimates the corresponding semantics-level Shapley interaction value for $< \mathbf{h}_i^I, \mathbf{h}_j^T >$ along with the uncertainty $\sigma$. The architectures of the three models are consistent with their token-level implementations.

# E  Pre-Training and Evaluation Details

## E.1  Pre-Training Dataset Details

As recent works [6, 14, 18] have shown that pre-training models can obtain great performance gain by scaling up the dataset, we construct a large-scale dataset, which consists of 240 million image-text pairs and covers a broad set of visual concepts. Concretely, we elaborate more details in the following.

**Raw image-text pair collection.** We first harvest large-scale noisy image-text pairs from the web and design multiple filtering rules to improve the quality of the web data.

**Image-based filtering.** Following ALIGN [6], we remove pornographic images and keep only images where both dimensions are larger than 200 pixels. Also, we remove the images whose aspect ratio is larger than 10. To prevent from leaking testing data, we remove the images that appear in all downstream evaluation datasets (e.g., MSCOCO, Flickr30K).

**Text-based filtering.** We remove the repeated captions and keep only English texts. The texts that are shorter than 3 words or longer than 100 words are discarded. As ALIGN [6], we also remove the texts that contain any rare token (outside of 100 million most frequent unigrams and bigrams from the raw dataset).

**Joint image-text filtering.** Although the above filtering rules have filtered out many noisy data, it is hard to detect the mismatched image-text pairs, where the texts do not accurately describe the visual content of the images, resulting in undesirable noisy signals to vision-language pre-training. Inspired by BLIP [9], we train a discriminator as a filtering model to predict whether the text is matched to the image. Specifically, the filtering model consists of an image encoder and an image-grounded text encoder, which takes the cross-attention to fuse image features and text features. The filtering model is trained on CC12M dataset using image-text contrastive loss and image-text matching loss.

## E.2  Evaluation Details

**Zero-Shot Image-Text Retrieval.** We evaluate the zero-shot performance of LOUPE on the image-text retrieval task over the widely used Flickr30K [13] and MSCOCO [11] datasets. The image-text retrieval consists of two subtasks: image-to-text retrieval and text-to-image retrieval, where a model is required to identify an image from candidates given a caption describing its content, or vice versa. The MSCOCO dataset consists of 123,287 images, and each image is aligned with five captions. The Flickr30K dataset contains 31,783 images and five captions for each image. Following previous works [6, 18], we evaluate the zero-shot performance on the 1K and 5K test sets of Flickr30K and MSCOCO, respectively. We take the final representation of `[CLS]` tokens as the global representations of images and texts, and use them to measure the image-text similarity. We first compute the similarity scores for all image-text pairs. Then, we take the top-K candidates for ranking and report the top-K retrieval results.

**Zero-Shot Transfer to Object Detection.** Without any fine-tuning, we evaluate the zero-shot transfer performance of LOUPE on the object detection task [16] over the COCO [11] and PASCAL VOC [4] datasets. For the COCO Objects dataset, we use their 2017 validation split for evaluation. Previous zero-shot object detection models [2, 15, 24] follow the split proposed by [2], which consists of 48

Table 2: Image captioning evaluation results on COCO "Karpathy" test split.

| | Image Captioning | | | |
| | BLEU@4 | METEOR | CIDEr | SPICE |
|---|---|---|---|---|
| VLP [23] | 36.5 | 28.4 | 117.7 | 21.3 |
| OSCAR$_{large}$ [10] | 37.4 | 30.7 | 127.8 | 23.5 |
| VinVL$_{large}$ [21] | 38.5 | 30.4 | 130.8 | 23.4 |
| BLIP$_{ViT-L}$ [9] | 40.4 | - | 136.7 | - |
| LEMON$_{large}$ [5] | 40.6 | 30.4 | 135.7 | 23.5 |
| **LOUPE** | **40.9** | **31.5** | **137.8** | **24.3** |

base classes and 17 novel classes. They train models on base classes and evaluate models on novel classes. Differently, we directly evaluate the zero-shot transfer performance on both the base and novel classes, without fine-tuning on the base classes. Totally, we evaluate models on 4,836 test images that contain 33,152 instances of 65 classes. PASCAL VOC is a widely used object detection dataset, which contains 20 object classes. For PASCAL VOC, we evaluate models on 9657 instances of 5072 images. To complete object detection, we first use the region generation module to generate a set of candidate regions and then use prompt text (*i.e., an image of [object class name].*) to expand each detection label to a sentence. Next, we encode sentences for each object class by the learned text encoder and measure their similarity with the candidate regions as the classification scores. Following most zero-shot object detection methods, we use mean Average Precision (mAP) at IoU of $\{0.3, 0.5\}$ as evaluation metrics.

**Zero-Shot Transfer to Visual Grounding.** Visual grounding [19] (also known as phrase localization and referring expression comprehension) aims to locate a specific visual region of the input image, according to the language referring expression. Visual grounding can be seen as generalized object detection, where the pre-defined class labels are replaced by language referring expression sentences. Without any fine-tuning, we evaluate the zero-shot transfer performance of LOUPE on the visual grounding task over the RefCOCO [19] and RefCOCO+ [19] datasets. These two datasets are collected by the ReferitGame [7], where a player is asked to write a language expression to refer to a specific object in the image, and another player is required to locate the target object given the image and the referring expression. RefCOCO dataset consists of 142,209 refer expressions for 50,000 objects in 19,994 images, which is split into train (120,624 expressions), val (10,834 expressions), testA (5,657 expressions), testB (5,095 expressions) sets. The images in testA set involve multiple persons and the images in testB set involve multiple objects. RefCOCO+ dataset consists of 141,564 expressions for 49,856 objects in 19,992 images, which is split into train (120,191 expressions), val (10,758 expressions), testA (5,726 expressions), testB (4,889 expressions) sets. We report the zero-shot transfer performance on the val, testA, and testB sets of both datasets.

# F More Experiment Results on Vision-Language Generation Task

To further validate the generalization ability of the learned cross-modal representations by our LOUPE, we adapt the pre-trained LOUPE to vision-language generation task, *i.e.*, image captioning [1]. Image captioning is the task of describing images with natural languages, which requires models to identify and describe the fine-grained semantics of images. The input images are encoded by the learned image encoder. As BLIP [9], we train an image-grounded text decoder which shares the feed forward layers with the learned text encoder and adopts cross-attention to attend to the image features. The text decoder is trained with a language modeling loss to generate captions according to the images.

We evaluate the image captioning performance on the MSCOCO [11] dataset, which is split into train (113, 287 images), val (5,000 images), "Karpathy" test split (5,000 images). Each image has 5 captions. We use the train split to train the image-grounded text decoder and report the performance on the public "Karpath" 5k test split. Following standard metrics, we use BLEU@4, METEOR, CIDEr, and SPICE as evaluation metrics. We compare our LOUPE model with recent vision-language pre-training generation models [5, 9, 10, 21, 23]. All methods are fine-tuned with cross-entropy loss only, without CIDEr optimization. As shown in Table 2, our LOUPE achieves competitive performance on all metrics, which verifies the strong generalization ability of our model on downstream vision-language generation tasks.

Table 3: Further ablation results (R@1) with respect to different image encoders.

| | | Image Encoder | Flickr30K | | MSCOCO | |
|---|---|---|---|---|---|---|
| | | | image-to-text | text-to-image | image-to-text | text-to-image |
| 1 | ALIGN [6] | EfficientNet | 88.6 | 75.7 | 58.6 | 45.6 |
| 2 | FILIP [18] | ViT-L | 89.8 | 75.0 | 61.3 | 45.9 |
| 3 | CLIP [14] | ViT-L | 88.0 | 68.7 | 58.4 | 37.8 |
| 4 | CLIP* | Swin-L | 88.7 | 74.3 | 59.3 | 46.2 |
| 5 | **LOUPE** | Swin-L | **90.5** | **76.3** | **62.3** | **50.1** |

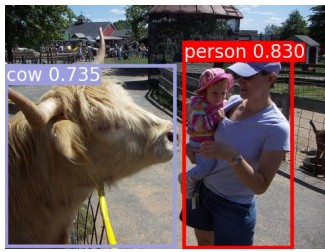 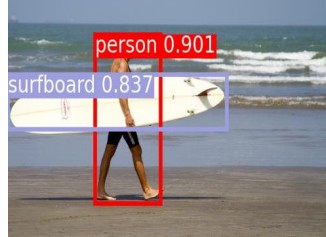 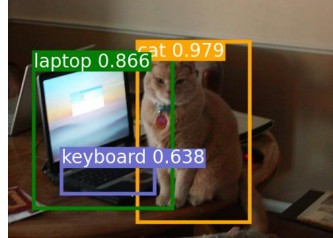

Figure 1: Qualitative examples of object detection on COCO Objects dataset.

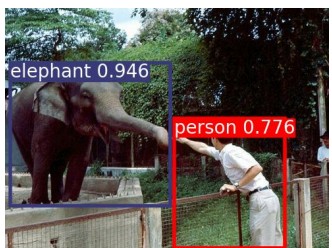 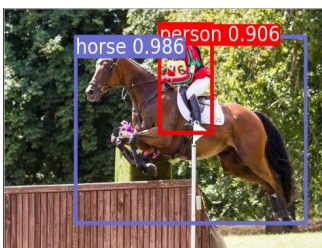 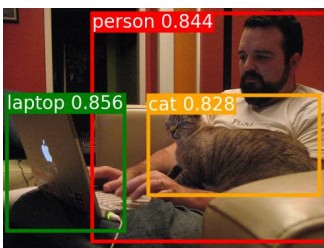

Figure 2: Qualitative examples of object detection on PASCAL VOC dataset.

# G    Further Analysis on the Image Encoder

In our work, we adopt the Swin-L [12] as our image encoder due to the following considerations. (1) The shifted windowing scheme of Swin Transformer achieves linear computational complexity with respect to image size, which is more efficient than ViT [3]. This merit is particularly beneficial to the vision-language pre-training as we need to process large-scale images (240M). (2) The hierarchical architecture of Swin Transformer is more flexible to model semantic regions at various scales.

To further verify the performance gain from our proposed fine-grained semantically aligned vision-language pre-training framework, we implement a variant version of CLIP that adopts Swin-L as the image encoder (Row 4 in Table 3), using the same training dataset as our LOUPE. It can also be viewed as the backbone of our LUOPE (without optimization from our token-level and semantics-level Shapley interaction modeling). As shown in Table 3, comparing CLIP* with CLIP, the Swin-L image encoder does bring some improvements over CLIP. However, there is still a clear performance gap between CLIP* and our LOUPE. With the same architecture, our LOUPE has 2.68 points higher average R@1 than the CLIP* over two datasets. This further verifies that the main performance gain comes from our proposed fine-grained semantically aligned vision-language pre-training framework. Notably, we observe that the text-to-image retrieval of our implementation is obviously higher than CLIP. This phenomenon has also been confirmed by [6, 18] (see Row 1 and Row 2 in Table 3). We suppose that it might be caused by some training details or the dataset collection of CLIP.

# H    More Qualitative Examples on Object Detection and Visual Grounding

For a more intuitive view of the performance of our model on object detection and visual grounding, we visualize more qualitative examples. Concretely, Figure 1 and Figure 2 show more object detection examples on the COCO [11] and PASCAL VOC [4] datasets. Figure 3 and Figure 4 show more visual grounding examples on the RefCOCO [19] and RefCOCO+ [19] datasets.

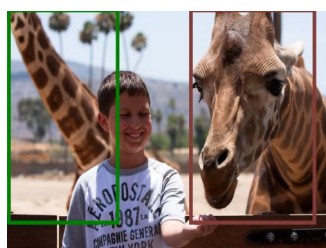 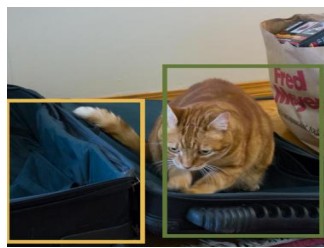 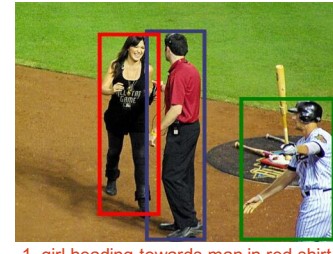

1. giraffe neck on the left
2. giraffe head near boy

1. left part of suitcase
2. part of suitcase that cat is sitting in

1. girl heading towards man in red shirt
2. man in red          3. player

Figure 3: Qualitative examples of visual grounding on RefCOCO dataset.

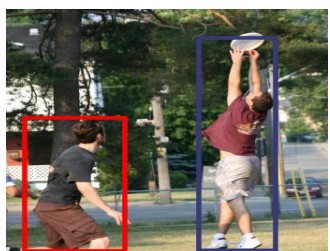 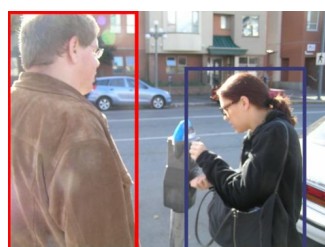 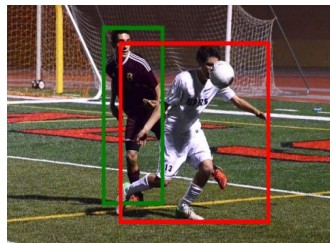

1. black shirt player
2. a man catching a frisbee

1. man with a brown jacket
2. woman

1. white clothing guy
2. black shirt guy

Figure 4: Qualitative examples of visual grounding on RefCOCO+ dataset.

Table 4: Linear probing performance (top-1 accuracy) over 11 datasets.

|  | CIFAR10 | Food101 | StanfordCars | SUN397 | Flowers102 | Country211 | FER2013 | Aircrafts | OxfordPets | Caltech101 | ImageNet |
|---|---|---|---|---|---|---|---|---|---|---|---|
| CLIP | **98.0** | 95.2 | 90.9 | 81.8 | 99.2 | 46.4 | **72.9** | 69.4 | 95.1 | 96.5 | 83.9 |
| **LOUPE** | 97.6 | **96.0** | **92.1** | **82.6** | **99.5** | **49.3** | 70.7 | **80.2** | **95.5** | **97.5** | **85.7** |

Table 5: Comparison of training cost and architecture parameters.

|  | Pre-Training Image-Text Pairs | Parameters | GPUs | Days | GPU Days |
|---|---|---|---|---|---|
| CLIP | 400M | 425M | 256 V100 | 12 days | 3072 |
| ALIGN | 1800M | 820M | 1024 TPUv3 | - | - |
| FILIP | 340M | 417M | 192 V100 | 24 days | 4608 |
| **LOUPE** | 240M | 226M | 128 V100 | 20 days | 2560 |

# I   Linear Probing Evaluation

In this section, we evaluate the linear probing performance of our LOUPE on image classification. Following the same evaluation setting as CLIP [14], we freeze the whole backbone network and only fine-tuning the last linear classification layer, which takes the `[CLS]` token as input. We report the linear probing performance over 11 datasets in Table 4. Our LOUPE outperforms CLIP with average improvement of 1.6%. Notably, on ImageNet, the largest dataset among 11 datasets, our LOUPE surpasses CLIP by 1.8%.

# J   Training Efficiency Discussion

Although our proposed Shapley interaction modeling increases the training time per iteration, it enables our model to converge with fewer total iterations by encouraging our model to learn fine-

grained region-phrase alignment beyond coarse image-text alignment. As shown in Table 5, our LOUPE achieves the best performance while using relatively small GPU days (128 GPUs × 20 days).

Indeed, the proposed Shapley interaction modeling increases the training time per iteration, but it enables our model to learn fine-grained region-phrase alignment from raw image-text pairs without any object-level human annotations. Our LOUPE can be used as a zero-shot object detector without any fine-tuning. Compared with the expensive cost of human annotations, the increased training time might be acceptable. Meanwhile, manual annotations for extremely diverse object categories in the real world are unscalable and even impossible while our model demonstrates a promising alternative, that is, learning fine-grained semantics from raw texts about images, which are easily available and contain a broader set of visual concepts. For example, the right case of Figure 4 in the main paper shows that LOUPE successfully recognizes the leash region and aligns it with the "a leash" phrase. Note that the "leash" category has never appeared in any existing object detection datasets.

On the other hand, our method is much more efficient than methods that rely on off-the-shelf object detectors (e.g., Faster R-CNN) to extract visual features. Recent studies [8, 18] have noticed that extracting visual features using object detectors greatly slows down the training (about 20 FPS per GPU) and requires more GPU memory. Thus, our model avoids such a heavy burden while being able to identify semantic-rich visual regions without any pre-training detectors or human annotations.

| Methods | Coarse-grained image-text alignment | Fine-grained region-phrase alignment | Ways to learn fine-grained region-phrase alignment |
|---|:---:|:---:|:---:|
| CLIP, ALIGN, DeCLIP | ✓ | - | - |
| ImageBERT, UNITER, FILIP, ViLT, ALBEF | ✓ | ✓ | Implicit supversion signals from end-to-end training (*e.g.*, Image-Text Contrastive loss) |
| GLIP, X-VLM, RegionCLIP | ✓ | ✓ | Human bounding-box annotations and supervised pre-trained Region Proposal Network |
| **LOUPE** | ✓ | ✓ | Explicit alignment information quantified by game-theoretic interactions |

Figure 5: Comparison of the LOUPE with existing methods.

# K    Detailed Discussion with Some Related Works

In this section, we first provide comparison table to highlight key differences of our LOUPE with various methods. Then, we provide a detailed discussion with three recent works (*i.e.*, FILIP [18], RegionCLIP [22], X-VLM [20]), which also investigate fine-grained semantic alignment.

We highlight key differences in Figure 5. Our LOUPE differs as it explicitly learns fine-grained region-phrase alignment from the novel perspective of game-theoretic interactions, without resorting to any object-level human annotations and pre-trained Region Proposal Network (RPN). Notably, the human bounding-box annotations are usually limited to the pre-defined object categories, and the RPN can only detect regions belonging to the pre-defined categories of pre-training object detection datasets. Thus, the methods that use human bounding-box annotations or pre-trained RPN usually

suffer from detecting novel objects beyond the pre-defined categories while LOUPE learns from large-scale raw image-text pairs, which are more scalable and contain a broader set of visual concepts.

Compared with FILIP, the superiorities of using Shapley Interaction modeling are mainly three-fold: **1)** We suppose that directly computing token-wise alignment between every patch token and word token is not efficient and meaningful because an individual word token or patch token might not contain complete semantics. A semantic-rich phrase (e.g., "a girl in a blue coat") usually consists of multiple words, and its corresponding visual region is composed of multiple patches. Also, some words (e.g., "is", "the") and patches (e.g., background pixels) are not meaningful. Based on this insight, our LOUPE differs as we first propose token-level Shapley interaction modeling to aggregate patches into semantic-meaningful regions, and then introduce semantics-level Shapley interaction modeling to explicitly model the fine-grained semantic alignment between semantic-meaningful regions and phrases. **2)** Although FILIP computes token-wise similarity to simulate the fine-grained alignment, it can only learn implicit alignment from the supervision of image-text contrastive loss, lacking training signals to explicitly encourage semantic alignment between visual regions and textual phrases. In contrast, our Shapley interaction modeling provides explicit supervision signals (e.g., the alignment matrices visualized in Figure 4) to learn the fine-grained alignment. The consistently superior performance of our LOUPE than FILIP over all metrics also demonstrates the benefit of explicit fine-grained alignment learning. **3)** FILIP can not be directly applied to object detection and visual grounding through implicit token-wise alignment learning while our LOUPE can immediately transfer to these tasks without any fine-tuning. It is because the proposed Shapley interaction modeling enables our model to identify semantic regions and align these regions with language. As shown in Table 2, without any bounding-box annotations and fine-tuning, our LOUPE achieves competitive performance across four object detection and visual grounding benchmarks.

Our LOUPE is different from RegionCLIP in the following aspects: **1)** RegionCLIP uses pre-trained Region Proposal Network (RPN) to detect regions in images. However, RPN is usually pre-trained on pre-defined object categories (e.g., 80 classes for MSCOCO), which can not cover extensive categories of objects in the large-scale pre-training dataset. Furthermore, since the RPN casts excessive demand on memory and computation, existing methods (i.e., RegionCLIP) usually fix the parameters of RPN and regard region detection as pre-processing step, disconnected with vision-language pre-training. Thus, the performance of RegionCLIP is also restricted by the quality of the RPN. In contrast, our LOUPE learns to identify semantic regions of images by token-level Shapley interaction modeling, which is more scalable and enables our LOUPE to learn a broader set of visual concepts from large-scale pre-training dataset. **2)** RegionCLIP constructs a pool of object concepts from image-text corpus and aligns visual regions with these concepts. These concepts are usually individual nouns (e.g., boy, kite, bus). In contrast, our LOUPE focuses on phrases that involve rich context (e.g., "a boy running on the grass"). By aligning visual regions with phrases that contain rich semantic context, our LOUPE can learn a boarder set of visual concepts (e.g., objects, actions, relations) from the large-scale pre-training dataset.

As for X-VLM, the main differences lie in three-fold: **1)** X-VLM is trained on well-annotated datasets, where regions with bounding-box annotations are provided and each of them is associated with a description text. Such a manner is time-consuming and hard to scale to larger raw image-text data from the web. Our LOUPE differs as we are trained on noisy image-text pairs from the Internet. **2)** X-VLM takes ground-truth regions as input and is trained to predict the bounding-box supervised by the regression loss on the ground-truth coordinates. In contrast, our LOUPE learns to identify semantic regions of images without such strong supervision signals from human annotations. **3)** X-VLM has ground-truth alignment information between regions and their corresponding description texts, which provide strong supervision signals for region-text matching. By comparison, our LOUPE learns the fine-grained region-phrase alignment from game-theoretic interactions.