# OpenReview forum: "Fine-Grained Semantically Aligned Vision-Language Pre-Training"
_NeurIPS.cc/2022/Conference — NeurIPS 2022 Accept_

### Official Review · Reviewer_7KYj · 2022-07-06

**Rating:** 6
**Confidence:** 3
**Soundness:** 3 good
**Presentation:** 3 good
**Contribution:** 3 good

**Summary:**

This paper focuses on learning fine-grained vision-language alignment under the widely used contrastive-based pre-training paradigm. The authors first point out that previous pre-training frameworks, *e.g.,* CLIP, only models global image-text alignment while neglecting the fine-grained semantic features. Alignments though can be achieved by interacting between modalities with the help of certain mechanisms like cross-attention, it lacks explicit and strong supervision to encourage meaningful patch-phrase correspondence, which proved to be quite useful for reasoning downstream tasks. The authors address the limitation by explicitly matching phrases in the text with regions of patches in the image. Different from previous approaches, the authors design light modules to achieve patch-to-patch and region-to-phrase alignment by introducing supervisions derived based on the Shapley value in game theory. Practically, Shapley interaction values are used as the criteria to estimate whether two patches belong to the same region and whether an image region corresponds to a certain phrase. In addition, to address the training efficiency problem when computing Shapley values, an additional estimator is adopted to predict the true values. Extensive experiments show the zero-shot ability of the proposed model on retrieval tasks, detection tasks, and visual grounding tasks. Ablative studies on the new losses and Shapley value approximation module are conducted. Limitations and social impacts are also discussed.

Overall, this paper shares an interesting idea by incorporating game theory mechanisms in the process of text/image alignments. This kind of supervision is free of pre-trained detection/segmentation models and does not require sophisticated text/image labels. The proposed model shows significant improvements on the zero-shot transfer ability and outperforms baseline approaches by a large margin.

**Questions:**

I have stated my major concerns in the **Weaknesses** section. The following are some detailed questions and suggestions.

+ How is the score function in the game defined? Does it need to be carefully designed?
+ The alignment between image regions and text phrases implicitly involves the concept of object. This may lead to similar limitations to the previous works that relied on pre-trained object detectors, and hard to generalize to object-free inputs. Also, according to line185, all regions are constrained within a scale of K patches. I wonder how does the model deal with objects (concepts) of different scale, *e.g.,* apple and sky?
+ Line 230. How long do we need to train the UNSIL module?
+ I am confused about Eq.(12). The authors claim the loss function is derived from the regression loss function. Could the authors provide a detailed derivation process? From my perspective the second term $\beta_2 \ \sigma$ should be $\beta_2 \ {\rm log}(\sigma)$.
+ I am also concerned about the training time shown in table 3. It seems like the additional cost is still huge even with the approximation module. I wonder if the authors can provide a training cost-performance trade-off compared with other works? Some qualitative analysis would also do the trick.
+ If possible, I would like to see a comparison between the proposed model and some missing related works, like RegionCLIP and X-VLM that were mentioned above. These approaches share similar insight and can also perform zero-shot transfer to several downstream tasks.

**Limitations:**

The authors have adequately addressed the limitations and social impacts in the paper.

**Strengths And Weaknesses:**

+ Strengths
  + This paper is well organized and easy to follow. The authors first give a brief introduction on the concept of Shapley values and Shapley interaction values and elaborate on how it works under the framework of multi-modal pre-training. The derivation seems to be sound and easy to understand. The main results and ablative studies help to better understand the proposed model.
  + The idea in this paper is quite interesting and seems novel to me. The idea of aligning image regions with text phrases has been explored in previous works, while most of them rely on additional label information and fail to provide strong supervision. Introducing Shapley value in the aligning process sounds reasonable and practically shows promising results.

+ Weaknesses
  + Some missing related works. Similar to this paper, there are also works that attempt to address the problem of missing fine-grained information. For example, RegionCLIP[1] also finds regions in the image and aligns them with text phrases. Off-the-shelf language parsers are similarly considered. Also, RegionCLIP is able to perform zero-shot object detection tasks. Another work, X-VLM[2], shares a similar insight. I believe these works are more related to the paper, and a detailed discussion on the connections and differences with these works is suggested.
  + Training efficiency. I understand that the authors use a sub-network to predict the real Shapley value to save excessive computation costs. Nevertheless, the training time in table 3 (comparing line 1 and line 4) shows that the proposed approach still involves 60~70 training time. This can be a great burden, especially for large-scale pre-training.
  + Please see the **Questions** section for some detailed concerns.

[1] Zhong, Yiwu, et al. "Regionclip: Region-based language-image pretraining." Proceedings of the IEEE/CVF Conference on Computer Vision and Pattern Recognition. 2022.
[2] Zeng, Yan, Xinsong Zhang, and Hang Li. "Multi-Grained Vision Language Pre-Training: Aligning Texts with Visual Concepts." arXiv preprint arXiv:2111.08276 (2021).

---

> ### Author Response · Authors · 2022-08-02
> **Responses to Reviewer 7KYj (Part 1)**
>
> We sincerely appreciate the reviewer for the constructive and insightful feedback. We are encouraged that the reviewer finds our idea is quite interesting and novel. We will explain your concerns point by point.
>
> **Q1: Some missing related works. Similar to this paper, there are also works that attempt to address the problem of missing fine-grained information. For example, RegionCLIP[1] also finds regions in the image and aligns them with text phrases. Off-the-shelf language parsers are similarly considered. Also, RegionCLIP is able to perform zero-shot object detection tasks. Another work, X-VLM[2], shares a similar insight. I believe these works are more related to the paper, and a detailed discussion on the connections and differences with these works is suggested.**
>
> **A1:** Thanks for the nice suggestion. Our LOUPE is different from RegionCLIP in the following aspects:
>
> 1. **RegionCLIP** uses pre-trained Region Proposal Network (RPN) to detect regions in images. However, RPN is usually pre-trained on pre-defined object categories (*e.g.*, 80 classes for MSCOCO), which can not cover extensive categories of objects in the large-scale pre-training dataset. Furthermore, since the RPN casts excessive demand on memory and computation, existing methods (*i.e.*, RegionCLIP) usually fix the parameters of RPN and regard region detection as pre-processing step, disconnected with vision-language pre-training. Thus, the performance of RegionCLIP is also restricted by the quality of the RPN. In contrast, our **LOUPE** learns to identify semantic regions of images by token-level Shapley interaction modeling, which is more scalable and enables our LOUPE to learn a broader set of visual concepts from large-scale pre-training datasets. For example, as shown in the right case of Figure 4, LOUPE successfully recognizes the leash region and aligns it with the “a leash” phrase. Note that the “leash” category has never appeared in any existing object detection datasets.
> 2. **RegionCLIP** constructs a pool of object concepts from the image-text corpus and aligns visual regions with these concepts. These concepts are usually individual nouns (*e.g.*, boy, kite, bus). In contrast, our **LOUPE** focuses on phrases that involve rich context (*e.g.*, "a boy running on the grass"). By aligning visual regions with phrases that contain rich semantic context, our LOUPE can learn a boarder set of visual concepts (*e.g.*, objects, actions, relations) from the large-scale pre-training dataset.
>
> As for X-VLM, the main differences lie in three-fold:
>
> 1. **X-VLM** is trained on well-annotated datasets, where regions with bounding-box annotations are provided and each of them is associated with a description text. Such a manner is time-consuming and hard to scale to larger raw image-text data from the web. Our **LOUPE** differs as we are trained on noisy image-text pairs from the Internet.
> 2. **X-VLM** takes ground-truth regions as input and is trained to predict bounding boxes supervised by the regression loss on the ground-truth coordinates. In contrast, our **LOUPE** learns to identify semantic regions of images without such strong supervision signals from human annotations.
> 3. **X-VLM** has ground-truth alignment information between regions and their corresponding description texts, which provide strong supervision signals for region-text matching. By comparison, our **LOUPE** learns the fine-grained region-phrase alignment from game-theoretic interactions.
>
> We will include these discussions in the next version according to your nice suggestion.

---

> > ### Author Response · Authors · 2022-08-02
> > **Responses to Reviewer 7KYj (Part 2)**
> >
> > **Q2: Training efficiency. I understand that the authors use a sub-network to predict the real Shapley value to save excessive computation costs. Nevertheless, the training time in table 3 (comparing line 1 and line 4) shows that the proposed approach still involves 60~70 training time. This can be a great burden, especially for large-scale pre-training.**
> >
> > **A2:** Thanks for raising this concern. Although our proposed Shapley interaction modeling increases the training time of per iteration, it enables our model to converge with fewer total iterations by encouraging our model to learn fine-grained region-phrase alignment beyond coarse image-text alignment. As you nicely suggested in the **Questions** section (Q7), we compare the training cost and performance of our LOUPE with other works in the following table. As shown in the following table, our LOUPE achieves the best performance while using relatively small GPU days (128 GPUs $\times$ 20 days).
> >
> > | Method    | &nbsp; &nbsp; &nbsp; GPUs    | Training Time | Flickr30K I2T | Flickr30K T2I | MSCOCO I2T | MSCOCO T2I |
> > | :-------- | :--------: | :-----------: | :-----------: | :-----------: | :--------: | :--------: |
> > | CLIP      |  256 V100  |    12 days    |     88.0      |     68.7      |    58.4    |    37.8    |
> > | ALIGN     | 1024 TPUv3 |       -       |     88.6      |     75.7      |    58.6    |    45.6    |
> > | FILIP     |  192 V100  |    24 days    |     89.8      |     75.0      |    61.3    |    45.9    |
> > | **LOUPE** |  128 V100  |    20 days    |   **90.5**    |   **76.3**    |  **62.3**  |  **50.1**  |
> >
> > Furthermore, our proposed Shapley interaction modeling:
> >
> > 1. enables our model to perform object detection in a zero-shot manner, avoiding expensive and time-consuming human annotations;
> >
> > 2. avoids using off-the-shelf object detectors (*e.g.*, Faster R-CNN) to extract visual features. Recent studies [A, B] have noticed that extracting visual features using object detectors greatly slows down the training (about 20 FPS per GPU) and requires more GPU memory.
> >
> > [A] ViLT: Vision-and-Language Transformer Without Convolution or Region Supervision. Kim et al. ICML 2021.
> >
> > [B] Filip: Fine-Grained Interactive Language-Image Pre-training. Yao et al. ICLR 2022.
> >
> > **Q3: How is the score function in the game defined? Does it need to be carefully designed?**
> >
> > **A3:** Thanks for your question. As you correctly understand, the score function needs to be carefully designed with the goal of reﬂecting the contribution of semantic regions and phrases. For token-level Shapley interaction modeling, the score function is defined as the global similarity between images and texts. To compute $v_1(S)$, we keep tokens in $S$ and mask tokens in $X \setminus S$ to zeros. Therefore, only tokens in $S$ contribute to the score function. For semantics-level Shapley interaction modeling, the score function is formulated as a carefully-designed fine-grained similarity score, which computes similarity based on region-phrase alignment scores. For more details, we respectfully refer the reviewer to Line 197-207 of the original paper. We will further clarify this in revision.
> >
> > **Q4: The alignment between image regions and text phrases implicitly involves the concept of object. This may lead to similar limitations to the previous works that relied on pre-trained object detectors, and hard to generalize to object-free inputs. Also, according to line185, all regions are constrained within a scale of K patches. I wonder how does the model deal with objects (concepts) of different scale, e.g., apple and sky?**
> >
> > **A4:** (1) Thanks for raising an important point. As mentioned in the differences with RegionCLIP, our LOUPE focuses on semantically rich phrases, which might contain diverse context. For example, we will extract the phrase "a man drinking wine alone" from the sentence "A woman looks at a man drinking wine alone". This phrase involves not only the object "man" and "wine", but also the action "drinking". Therefore, our LOUPE can learn a boarder set of visual concepts (*e.g.*, objects, actions, relations) from the large-scale image-text data.
> >
> > (2) Thanks for pointing out a confusing notation. K is not a fixed hyper-parameter. Different regions might have different numbers of patches, which is determined by the scale of the predicted bounding boxes. Now, we revise the notation as $\mathcal{R}_i = \\{ \mathbf{x}\_{i, k}^I  \\}\_{k=1}^{K_i}$, where $K_i$ is the number of patches for $\mathcal{R}_i$. We will include this revision and clarify this in the next version.
> >
> > **Q5: Line 230. How long do we need to train the UNSIL module?**
> >
> > **A5:** Thanks for your question. It takes about 20 hours to warm-start the UNSIL module.

---

> > > ### Author Response · Authors · 2022-08-02
> > > **Responses to Reviewer 7KYj (Part 3)**
> > >
> > > **Q6: I am confused about Eq.(12). The authors claim the loss function is derived from the regression loss function. Could the authors provide a detailed derivation process? From my perspective the second term  $\beta_2\sigma$ should be $\beta_2log(\sigma)$.**
> > >
> > > **A6:** Thanks for your question. We respectfully clarify our expression in Line 236. We do not mean that Eq. 12 is derived from a specific regression loss. Instead, we mean the form of Eq. 12 is a regression loss function, where we optimize the mean squared error between $\hat{\mathfrak{I}}$ and $\mathfrak{I}^*$. As you correctly point out, the second term is $log(\sigma)$ in original noisy label learning papers. Here, we replace $log(\sigma)$ with $\beta_2\sigma$ because we empirically find it is more numerically stable and can also achieve good performance. We will clarify this in revision.
> > >
> > > **Q7: I am also concerned about the training time shown in table 3. It seems like the additional cost is still huge even with the approximation module. I wonder if the authors can provide a training cost-performance trade-off compared with other works? Some qualitative analysis would also do the trick.**
> > >
> > > **A7:** Thanks for your constructive suggestion. Please refer to our reply to the Q2 of you. As you nicely suggested, we will include this discussion in the next version.
> > >
> > > **Q8: If possible, I would like to see a comparison between the proposed model and some missing related works, like RegionCLIP and X-VLM that were mentioned above. These approaches share similar insight and can also perform zero-shot transfer to several downstream tasks.**
> > >
> > > **A8:** Thanks for your suggestion. Although both RegionCLIP and X-VLM focus on fine-grained semantics, they  rely on pre-trained object detectors or manual bounding-box annotations. Therefore, they are limited to a closed set of object categories, which is pre-defined by the detectors or annotation labels. In contrast, our LOUPE can detect open-vocabulary categories of objects, without resorting to any human annotations or pre-trained detectors. Furthermore, our LOUPE can handle more diverse scenarios, such as zero-shot visual grounding. Based on the above analysis, we suppose that our LOUPLE might perform better on open-vocabulary object detection. We appreciate that the reviewer points out a meaningful and interesting direction for future research. In the future, we will attempt to compare our LOUPE with these methods in open-vocabulary object detection.

---

> > > > ### Comment · Reviewer_7KYj · 2022-08-04
> > > > **Responses to Authors**
> > > >
> > > > I would like to thank the authors for their timely responses. Most of my concerns are adequately addressed. The only remained suggestion is the comparison with more related baselines, which I believe would better demonstate the effectiveness of the work.
> > > >
> > > > In general, I believe this paper is an interesting touch towards learning fine-grained multi-modal relations, and the result seems to be promising. From my perpespective, the quality of this paper is above the acceptance threshold for NeurIPS2022.

---

> > > > > ### Author Response · Authors · 2022-08-04
> > > > > **Responses to Reviewer 7KYj (Part 4)**
> > > > >
> > > > > We appreciate the reviewer for the positive and insightful feedback. We quite agree with your suggestion and will explore it as an important future work. Also, inspired by your nice suggestion, we suppose that it is interesting and potential to enhance our LOUPE model by additional supervision from pre-trained detectors or human annotations, which might be able to work in a mutually enhanced way. Thanks for your insightful suggestion!

---

### Official Review · Reviewer_H9ch · 2022-07-10

**Rating:** 4
**Confidence:** 5
**Soundness:** 3 good
**Presentation:** 3 good
**Contribution:** 3 good

**Summary:**

This paper propose a fine-grained semantically aligned vision-language pre-training framework（LOUPE） from game-theoretic interactions. Experiments on image-text retrieval, object detection and visual grounding tasks demonstrate the effectiveness of LOUPE.


**Questions:**

1. The experiments are not sufficient. The model structures（both image encoder and text encoder) of different methods in Tabel 1 and Table 2 are inconsistent and the comparison in unfair.

2. Can plain Dual-Encoder be used for zero-shot classification? How is the result on ImageNet dataset?

3. Why are the loss weights of $L_{CMC}$, $L_{TSA}$ and $L_{FSA}$ set to 1:1:1?

4. Are public datasets, such as CC12M, included in the 240M dataset? In addition to the description part in Supplementary Material E, it is hoped that a more detailed introduction to the 240M dataset can be provided.

**Limitations:**

The authors have adequately addressed the limitations and potential negative societal impact of their work

**Strengths And Weaknesses:**

1. The motivation is clear. LOUPE propose Phrase-Region Semantic Alignment to achieve fine-grained vision-language pre-training.

2. The hybrid Shapley interaction learning strategy is interesting for me.

---

> ### Author Response · Authors · 2022-08-02
> **Response to Reviewer H9ch (Part 1)**
>
> We appreciate the reviewer for the valuable comments. Our response to the reviewer’s questions is as follows.
>
> **Q1: The experiments are not sufficient. The model structures（both image encoder and text encoder) of different methods in Tabel 1 and Table 2 are inconsistent and the comparison in unfair.**
>
> **A1:** (1) Thanks for raising this concern. Our text encoder is implemented by BERT-Small, which is consistent with most methods (*e.g.*, ALIGN uses BERT-Large, ALBEF uses BERT-Base, UNITER uses BERT). For the image encoder, we observe that it varies with different methods (*e.g.*, FILIP uses ViT-L, ALIGN uses EfficientNet, UNITER uses Faster R-CNN, X-VLM uses Swin). As discussed in Appendix G, in our work, we adopt the Swin-L as our image encoder due to the following considerations:
>
> 1. The shifted windowing scheme of Swin Transformer achieves linear computational complexity with respect to image size, which is more efficient than ViT. This merit is particularly beneficial to the vision-language pre-training as we need to process large-scale images (240M).
> 2. The hierarchical architecture of Swin Transformer is more ﬂexible to model semantic regions at various scales.
>
> (2) To further verify the real performance gain from our proposed fine-grained semantically aligned vision-language pre-training framework, we implement a variant version of CLIP that adopts Swin-L as the image encoder, using the same training dataset as our LOUPE. As shown in the following table, comparing CLIP* with CLIP, the Swin-L image encoder does bring some improvements over CLIP. However, there is still a clear performance gap between CLIP* and our LOUPE. With the same architecture, our LOUPE has 2.68 points higher average R@1 than the CLIP* over two datasets. This further verifies that the main performance gain comes from our proposed fine-grained semantically aligned vision-language pre-training framework. Notably, we observe that the text-to-image retrieval of our implementation is obviously higher than CLIP. This phenomenon has also been confirmed by [B, D] (see Row 1 and Row 2 in the table). We suppose that it might be caused by some training details or the dataset collection of CLIP. We refer the reviewer to Appendix G for more details.
>
> |           | Image Encoder | Flickr30K I2T | Flickr30K T2I | MSCOCO I2T | MSCOCO T2I |
> | :-------- | :-----------: | :-----------: | :-----------: | :--------: | :--------: |
> | ALIGN     | EfficientNet  |     88.6      |     75.7      |    58.6    |    45.6    |
> | FIILIP    |     ViT-L     |     89.8      |     75.0      |    61.3    |    45.9    |
> | CLIP      |     ViT-L     |     88.0      |     68.7      |    58.4    |    37.8    |
> | CLIP*     |    Swin-L     |     88.7      |     74.3      |    59.3    |    46.2    |
> | **LOUPE** |    Swin-L     |   **90.5**    |   **76.3**    |  **62.3**  |  **50.1**  |
>
> [B] Filip: Fine-Grained Interactive Language-Image Pre-training. Yao et al. ICLR 2022.
>
> [D] Scaling up visual and vision-language representation learning with noisy text supervision. Jia et al. PMLR 2021.
>
> **Q2: Can plain Dual-Encoder be used for zero-shot classification? How is the result on ImageNet dataset?**
>
> **A2:** Thanks for the constructive suggestion. Our answer is yes. As you nicely suggested, we add the zero-shot image classification and linear probing experiments over 11 datasets. For zero-shot image classification, our LOUPE outperforms CLIP with average improvement of 3.1%. Specifically, our LOUPE achieves 76.1% top-1 accuracy on ImageNet, surpassing CLIP by 0.8%. For linear probing evaluation, LOUPE achieves average improvement of 1.6% over CLIP. Specifically, LOUPE achieves 85.7% top-1 accuracy on ImageNet, surpassing CLIP by 1.8%. Please refer to our reply to the Q1 of Reviewer gkzN for more detailed results.

---

> > ### Author Response · Authors · 2022-08-02
> > **Response to Reviewer H9ch (Part 2)**
> >
> > **Q3: Why are the loss weights of $L_{CMC}, L_{TSA}$ and $L_{FSA}$ set to 1:1:1?**
> >
> > **A3:** Thanks for your question. We empirically find that it already performs well when we assign these losses the same weight without carefully tuning the weight hyper-parameter. Thanks for your suggestion, and we will explore different weight hyper-parameters in the future.
> >
> > **Q4: Are public datasets, such as CC12M, included in the 240M dataset? In addition to the description part in Supplementary Material E, it is hoped that a more detailed introduction to the 240M dataset can be provided.**
> >
> > **A4:** Thanks for your concern. Our pre-training dataset does not include any well-annotated public datasets, such as CC12M, COCO, and Visual Genome. As suggested, we elaborate more details in the following：
> >
> > 1. **Raw image-text pair collection.** We first harvest large-scale noisy image-text pairs from the web and design multiple filtering rules to improve the quality of the web data.
> > 2. **Image-based filtering.** Following ALIGN [D], we remove pornographic images and keep only images where both dimensions are larger than 200 pixels. Also, we remove the images whose aspect ratio is larger than 10. To prevent from leaking testing data, we remove the images that appear in all downstream evaluation datasets (*e.g.*, MSCOCO, Flickr30K).
> > 3. **Text-based filtering.** We remove the repeated captions and keep only English texts. The texts that are shorter than 3 words or longer than 100 words are discarded. As ALIGN [D], we also remove the texts that contain any rare token (outside of 100 million most frequent unigrams and bigrams from the raw dataset).
> > 4. **Joint image-text filtering.** Although the above filtering rules have filtered out many noisy data, it is hard to detect the mismatched image-text pairs, where the texts do not accurately describe the visual content of the images, resulting in undesirable noisy signals to vision-language pre-training. Inspired by BLIP [E], we train a discriminator as a filtering model to predict whether the text is matched to the image. Specifically, the filtering model consists of an image encoder and an image-grounded text encoder, which takes the cross-attention to fuse image features and text features. The filtering model is trained on CC12M dataset using image-text contrastive loss and image-text matching loss.
> >
> > We will add the details in the next version according to the reviewer's suggestion.
> >
> >
> >
> > [D] Scaling up visual and vision-language representation learning with noisy text supervision. Jia et al. PMLR 2021.
> >
> > [E] BLIP: Bootstrapping Language-Image Pre-training for Unified Vision-Language Understanding and Generation. Li et al. ICML 2022.

---

> > ### Comment · Reviewer_H9ch · 2022-08-03
> > **Reply to Response**
> >
> > 1）Did A1(2) use the full 240M data? How many GPUs did the CLIP* experiment use and how long did it take?
> >
> > 2）Can you provide results on the 11 datasets in A2 respectively? And how is the corresponding hyperparameter configuration?

---

> > > ### Author Response · Authors · 2022-08-03
> > > **Response to Reviewer H9ch (Part 3)**
> > >
> > > Many thanks for your prompt reply!
> > >
> > > **1) Did A1(2) use the full 240M data? How many GPUs did the CLIP\* experiment use and how long did it take?**
> > >
> > > **A1:** Thanks for your important concern. The CLIP* is also pre-trained on the full 240M dataset using 128 V100 GPUs, keeping almost the same pre-training details (*e.g.*, optimizer, learning schedule) as our LOUPE. The CLIP* takes about 22 days to train on 128 cards.
> > >
> > > **2) Can you provide results on the 11 datasets in A2 respectively? And how is the corresponding hyperparameter configuration?**
> > >
> > > **A2:** Thanks for your suggestion. We provide the detailed results in the following tables.
> > >
> > > i. Results (top-1 accuracy) of zero-shot image classiﬁcation over 11 datasets.
> > > |           | CIFAR10  | Food101  | StanfordCars |  SUN397  | Flowers102 | Country211 |
> > > | :-------- | :------: | :------: | :----------: | :------: | :--------: | :--------: |
> > > | CLIP      | **96.2** |   92.9   |     77.3     |   67.7   |    78.7    |    34.9    |
> > > | **LOUPE** |   95.9   | **94.3** |   **79.9**   | **69.8** |  **87.4**  |  **37.8**  |
> > >
> > > |           | FER2013  | Aircrafts | OxfordPets | Caltech101 | ImageNet |
> > > | :-------- | :------: | :-------: | :--------: | :--------: | :------: |
> > > | CLIP      | **57.7** |   36.1    |    93.5    |    92.6    |   75.3   |
> > > | **LOUPE** |   53.3   | **54.9**  |  **94.1**  |  **93.9**  | **76.1** |
> > >
> > >
> > >
> > > ii. Linear probing performance (top-1 accuracy) over 11 datasets.
> > > |           | CIFAR10  | Food101  | StanfordCars |  SUN397  | Flowers102 | Country211 |
> > > | :-------- | :------: | :------: | :----------: | :------: | :--------: | :--------: |
> > > | CLIP      | **98.0** |   95.2   |     90.9     |   81.8   |    99.2    |    46.4    |
> > > | **LOUPE** |   97.6   | **96.0** |   **92.1**   | **82.6** |  **99.5**  |  **49.3**  |
> > >
> > > |           | FER2013  | Aircrafts | OxfordPets | Caltech101 | ImageNet |
> > > | :-------- | :------: | :-------: | :--------: | :--------: | :------: |
> > > | CLIP      | **72.9** |   69.4    |    95.1    |    96.5    |   83.9   |
> > > | **LOUPE** |   70.7   | **80.2**  |  **95.5**  |  **97.5**  | **85.7** |
> > >
> > >
> > >
> > >
> > > For linear probing evaluation, we follow the same setting as CLIP. Specifically, we freeze the whole backbone model and use the final representation of the [CLS] token as the global image representation. Then, we train a linear classifier on the global image representation and report the top-1 accuracy for each dataset. The following table shows the hyperparameter configuration on these 11 datasets.
> > >
> > > | Image Size | Training Epochs | Batch Size | Optimizer | Learning Rate | Weight Decay |
> > > | :--------: | :-------------: | :--------: | :-------: | :-----------: | :----------: |
> > > |  224*224   |       100       |    512     |   AdamW   |  $ 3e^{-5}$   |     0.1      |
> > >
> > > We hope we have addressed all of your concerns. Waiting for your kind reply at any time. Thank you!

---

> > > > ### Comment · Reviewer_H9ch · 2022-08-04
> > > > **Reply to Response**
> > > >
> > > > I didn't find the description of linear probing configuration in CLIP paper, could you tell me where this part is? If I didn't miss something, CLIP uses logistic regression.

---

> > > > > ### Author Response · Authors · 2022-08-04
> > > > > **Response to Reviewer H9ch (Part 4)**
> > > > >
> > > > > Thanks for your prompt reply. As you said, CLIP uses a logistic regression classifier, which is a typical linear classifier. We follow the setting as claimed in Appendix A.3.Evaluation of CLIP paper. According to Appendix A.3.Evaluation of CLIP paper, we take the representations from the frozen image encode as input features and train a linear classifier, *i.e.*, logistic regression. For the further hyperparameter configuration of linear probing, it is not provided in both the CLIP paper and its official code implementation. Thus, we implement these hyperparameters as we reported in the table. We hope we have addressed your concerns. Waiting for your kind reply at any time. Thank you!

---

> ### Author Response · Authors · 2022-08-08
> **Response to Reviewer H9ch (Part 5)**
>
> Dear reviewer, we have tried to address your concerns in our earlier responses. If you have any additional questions or suggestions, we are very happy to discuss with you.

---

> > ### Author Response · Authors · 2022-08-09
> > **Response to Reviewer H9ch**
> >
> > Dear reviewer, do you have any further concerns or suggestions? We are very delighted to discuss with you.

---

> ### Author Response · Authors · 2022-08-09
> **Further response to Reviewer H9ch**
>
> Dear reviewer, since the discussion stage is about to end, do you have any major concerns or suggestions? We are happy to discuss with you.

---

### Official Review · Reviewer_gkzN · 2022-07-12

**Rating:** 6
**Confidence:** 4
**Soundness:** 3 good
**Presentation:** 4 excellent
**Contribution:** 3 good

**Summary:**

This paper proposed a fine-grained semantically aligned vision-language pre-training framework. Both visual token-level semantics alignment and phrase-region level semantics alignment are studied. To be more specific, the paper measures the shapley interaction of visual regions to text as a supervision signal for region generation module and visual backbone. Then the alignment between regions and phrases is also supervised by Shapley interaction between them. Further, an uncertainty-aware learning module is introduced to predict Shapley interaction to save the time cost of sampling-based estimation.

**Questions:**

Please refer to weaknesses.

**Ethics Review Area:**

["I don’t know"]

**Limitations:**

It's not guaranteed that the dataset they use doesn't contain unsuitable images, text, or personal information.

**Strengths And Weaknesses:**

Strengths:
1. The methodology is interesting and novel. It's essential to exploit the local correlation between image and text in contrastive pre-training. The idea of introducing Shapley interaction as supervision seems to fit the problem well.
2. LOUPE can outperform recent works trained with a similar scale of data in retrieval tasks. And the ablation is comprehensive.
3. The region generation model can also be used as a zero-shot object detector, which is quite impressive.

Weaknesses:
1. The evaluations on zero-shot image classification and linear probing are missing. Those are important to prove whether the learned visual backbone is strong and robust.  In CLIP, they reported the above two evaluations in 20+ datasets to prove the transferability of their visual backbone.
2. The training time needed is increased by 65% (1.17->1.93).
3. The dataset used seems to be a private dataset and it's hard to convince people that the comparison is fair enough. Although the scale of the proposed dataset is smaller, the quality is not compared and it matters a lot in CLIP-like pretraining. Also, it's not indicated whether the dataset will be released or not. If not, it's hard for people to reproduce this work.

================================
After rebuttal, weakness 1 and 2 have been alleviated. So I change the score.

---

> ### Author Response · Authors · 2022-08-02
> **Response to Reviewer gkzN (Part 1)**
>
> We sincerely thank you for your comprehensive comments and constructive advice. We will explain your concerns point by point.
>
> **Q1: The evaluations on zero-shot image classification and linear probing are missing. Those are important to prove whether the learned visual backbone is strong and robust. In CLIP, they reported the above two evaluations in 20+ datasets to prove the transferability of their visual backbone.**
>
> **A1:** Thanks for the constructive suggestion. Following the same setting as CLIP, we add the **zero-shot image classification** and **linear probing** experiments over 11 datasets and report the top-1 accuracy in the following tables. Moreover, we evaluate the transferability of  our LOUPE on **vision-language generation task**, *i.e.*, image captioning.
>
> (1) For zero-shot image classification, as shown in the following table, our LOUPE outperforms CLIP with average improvement of 3.1%. Notably, on ImageNet, the largest dataset among 11 datasets, our LOUPE surpasses CLIP by 0.8%. Also, we observe that LOUPE achieves substantial performance gains on several fine-grained image classification datasets (*i.e.*, Flowers102 and Aircrafts). It demonstrates the superiority of our LOUPE on fine-grained semantics understanding.
> |           | CIFAR10  | Food101  | StanfordCars |  SUN397  | Flowers102 | Country211 |
> | :-------- | :------: | :------: | :----------: | :------: | :--------: | :--------: |
> | CLIP      | **96.2** |   92.9   |     77.3     |   67.7   |    78.7    |    34.9    |
> | **LOUPE** |   95.9   | **94.3** |   **79.9**   | **69.8** |  **87.4**  |  **37.8**  |
>
> |           | FER2013  | Aircrafts | OxfordPets | Caltech101 | ImageNet |
> | :-------- | :------: | :-------: | :--------: | :--------: | :------: |
> | CLIP      | **57.7** |   36.1    |    93.5    |    92.6    |   75.3   |
> | **LOUPE** |   53.3   | **54.9**  |  **94.1**  |  **93.9**  | **76.1** |
>
> (2) For linear probing evaluation, as shown in the following table, our LOUPE outperforms CLIP with average improvement of 1.6%. Notably, on ImageNet, the largest dataset among 11 datasets, our LOUPE surpasses CLIP by 1.8%.
>
> |           | CIFAR10  | Food101  | StanfordCars |  SUN397  | Flowers102 | Country211 |
> | :-------- | :------: | :------: | :----------: | :------: | :--------: | :--------: |
> | CLIP      | **98.0** |   95.2   |     90.9     |   81.8   |    99.2    |    46.4    |
> | **LOUPE** |   97.6   | **96.0** |   **92.1**   | **82.6** |  **99.5**  |  **49.3**  |
>
> |           | FER2013  | Aircrafts | OxfordPets | Caltech101 | ImageNet |
> | :-------- | :------: | :-------: | :--------: | :--------: | :------: |
> | CLIP      | **72.9** |   69.4    |    95.1    |    96.5    |   83.9   |
> | **LOUPE** |   70.7   | **80.2**  |  **95.5**  |  **97.5**  | **85.7** |
>
>
> (3) Furthermore, we evaluate our LOUPE on vision-language generation task, *i.e.*, image captioning, to demonstrate the generalization ability of the learned cross-modal representations by our LOUPE. As shown in the following table, our LOUPE achieves competitive performance on all metrics, which verifies the strong generalization ability of our model on downstream vision-language generation tasks. We refer the reviewer to Appendix F for more details.
>
> | Method      |  BLEU@4  |  METEOR  |   CIDEr   |  SPICE   |
> | ----------- | :------: | :------: | :-------: | :------: |
> | VLP         |   36.5   |   28.4   |   117.7   |   21.3   |
> | OSCAR-Large |   37.4   |   30.7   |   127.8   |   23.5   |
> | VinVL-Large |   38.5   |   30.4   |   130.8   |   23.4   |
> | BLIP-ViT-L  |   40.4   |    -     |   136.7   |    -     |
> | LEMON-Large |   40.6   |   30.4   |   135.7   |   23.5   |
> | **LOUPE**   | **40.9** | **31.5** | **137.8** | **24.3** |

---

> > ### Author Response · Authors · 2022-08-02
> > **Response to Reviewer gkzN (Part 2)**
> >
> > **Q2: The training time needed is increased by 65% (1.17->1.93).**
> >
> > **A2:** Thanks for raising a concern about training efficiency.
> >
> > (1) Although our proposed Shapley interaction modeling increases the training time per iteration, it enables our model to converge with fewer total iterations by encouraging our model to learn fine-grained region-phrase alignment beyond coarse image-text alignment. As shown in the following table, our LOUPE achieves the best performance while using relatively small GPU days (128 GPUs $\times$ 20 days).
> >
> > | Method    | &nbsp; &nbsp; &nbsp; GPUs    | Training Time | Flickr30K I2T | Flickr30K T2I | MSCOCO I2T | MSCOCO T2I |
> > | :-------- | :--------: | :-----------: | :-----------: | :-----------: | :--------: | :--------: |
> > | CLIP      |  256 V100  |    12 days    |     88.0      |     68.7      |    58.4    |    37.8    |
> > | ALIGN     | 1024 TPUv3 |       -       |     88.6      |     75.7      |    58.6    |    45.6    |
> > | FILIP     |  192 V100  |    24 days    |     89.8      |     75.0      |    61.3    |    45.9    |
> > | **LOUPE** |  128 V100  |    20 days    |   **90.5**    |   **76.3**    |  **62.3**  |  **50.1**  |
> >
> > (2) Indeed, the proposed Shapley interaction modeling increases the training time per iteration, but it enables our model to learn fine-grained region-phrase alignment from raw image-text pairs without any object-level human annotations. As you nicely recognize, our LOUPE can be used as a zero-shot object detector without any fine-tuning. Compared with the expensive cost of human annotations, the increased training time might be acceptable. Meanwhile, manual annotations for extremely diverse object categories in the real world are unscalable and even impossible while our model demonstrates a promising alternative, that is, learning fine-grained semantics from raw texts about images, which are easily available and contain a broader set of visual concepts. For example, as shown in the right case of Figure 4, LOUPE successfully recognizes the leash region and aligns it with the “a leash” phrase. Note that the “leash” category has never appeared in any existing object detection datasets.
> >
> > (3) On the other hand, our method is much more efficient than methods that rely on off-the-shelf object detectors (*e.g.*, Faster R-CNN) to extract visual features. Recent studies [A, B] have noticed that extracting visual features using object detectors greatly slows down the training (about 20 FPS per GPU) and requires more GPU memory. Thus, our model avoids such a heavy burden while being able to identify semantic-rich visual regions without any pre-training detectors or human annotations.
> >
> > We respectfully hope the reviewer could reconsider the superiority and scalability brought from our LOUPE.
> >
> >
> >
> > [A] ViLT: Vision-and-Language Transformer Without Convolution or Region Supervision. Kim et al. ICML 2021.
> >
> > [B] Filip: Fine-Grained Interactive Language-Image Pre-training. Yao et al. ICLR 2022.
> >
> > **Q3: The dataset used seems to be a private dataset and it's hard to convince people that the comparison is fair enough. Although the scale of the proposed dataset is smaller, the quality is not compared and it matters a lot in CLIP-like pretraining. Also, it's not indicated whether the dataset will be released or not. If not, it's hard for people to reproduce this work.**
> >
> > **A3:** Thanks for the suggestion. As sufficient data is a prerequisite for vision-language pre-training, recent CLIP, ALIGN, and FILIP construct datasets with 400M, 1800M, and 340M image-text pairs, respectively. Since they are not publicly available, we also collect 240M noisy image-text pairs from the Internet. Note that we do not include any well-annotated datasets, such as CC12M, COCO, and Visual Genome. To facilitate future research and fair comparison, we will carefully review our collected dataset and consider releasing it in the future. Moreover, as we understand the cost of large-scale pre-training might be unaffordable for colleges and individual researchers, we plan to release the code and pre-trained model to promote more future research on downstream tasks and applications. In addition, we notice a recently released dataset LAION-400M [C], which makes it possible to fairly benchmark the performance of large-scale vision-language pre-training models. We would evaluate our model on this dataset as a future work.
> >
> >
> >
> > [C] LAION-400M: Open Dataset of CLIP-Filtered 400 Million Image-Text Pairs. Schuhmann et al. Arxiv: 2111.02114.

---

> ### Author Response · Authors · 2022-08-06
> **Response to Reviewer gkzN**
>
> Dear reviewer, we have tried to address your concerns in our earlier response. If you have any further questions or suggestions, we are very happy to discuss with you.

---

> > ### Comment · Reviewer_gkzN · 2022-08-08
> > **Response to rebuttal**
> >
> > Thanks the authors for those experiments and explanation. Most of my concerns are solved. As for pre-training dataset, I still hope to see a more fair comparison against other models on acceptable-scale public datasets (YFCC15M, LAION). I will raise my score.

---

> > > ### Author Response · Authors · 2022-08-08
> > > **Thank you for your acknowledgement!**
> > >
> > > Thanks for your positive and insightful feedback. We really appreciate your constructive review and your precious time. We quite agree with your suggestion of a further comparison against other models on acceptable-scale public datasets (YFCC15M, LAION). We will include it as an important future work.

---

### Official Review · Reviewer_yCwd · 2022-07-12

**Rating:** 6
**Confidence:** 3
**Soundness:** 3 good
**Presentation:** 2 fair
**Contribution:** 3 good

**Summary:**

The paper addresses the problem of vision-language pretraining, whose goal is to learn a strong backbone network transferable into downstream tasks such as object detection or visual grounding.
Thus, the paper proposes a framework for aligning between visual and textual tokens during training motivated by Shapley value from game theory.
Specifically, the model computes the Shapley values to determine the interaction between related image patches, which form objects or related image regions and noun phrases corresponding to object types in images.
The Shapley values for patches in image regions and pairs of image regions and noun phrases are used as soft pseudo-labels to train the alignment between textual and visual modalities.
The paper also introduces an approximation mechanism to reduce the complexity of Shapley value computation via sampling.
The paper evaluates the performances of Image-Text Retrieval, Object Detection, and Visual Grounding on MSCOCO and Flickr30k datasets.

------- After rebuttal -----

The rebuttal sufficiently addresses my concern as well as confirming my understanding.
Thus, I keep my rating and recommend the paper for acceptance.

**Questions:**

+ Please refer to the weakness section.


**Limitations:**

+ I believe the work doesn't seem to have any potential negative societal impact.
+ It would be interesting to investigate the bias from web data in future directions. Would the model propagate these biases from captioned images toward downstream tasks?

**Strengths And Weaknesses:**

#### Strength:
+ The formulation of semantic generation as well as semantic alignment under Shapley interaction is interesting and novel. Moreover, viewing weakly supervised visual-textual alignment in terms of game theory is a promising direction to explore.
+ Learning to predict Shapley more, as well as its uncertainty, seems to significantly reduce the training complexity.
+ The paper compares with appropriate baselines to demonstrate its effectiveness.

#### Weakness:
+ The technical section could be challenging to follow, especially for computer vision audiences without prior knowledge of Shapley value. The reviewer believes it would be very helpful if the paper could include a table to highlight key differences of the proposed work with various methods used for vision-language pretraining. If the reviewer understands correctly, the main distinction compared to other works would be the proposal of using Shapley interaction as soft pseudo labels for fine-grained image regions instead of using contrastive learning between image-caption pairs. However, this is not clear and emphasized enough in the paper.
+ The reviewer is unclear how Shapley value could be used as soft pseudo labels. For example, in line 180, the paper mentions that the game score $v_1$, which is used to compute the Shapley interaction, is the similarity between image and text. However, it appears that the paper does not mention any training phase for learning image-textual similarity. Then how would this similarity be computed correctly? Are there any pretraining phases or pretrained models employed for similarity computation?
+ The paper lacks convincing motivation on why using Shapley value instead of token-wise alignment (FILIP [42]) is beneficial. Although in lines 97-99, the paper mentions that FILIP has quadratic complexity, the proposed method also suffers from combinatorial complexity, which could be worse.
+ Although the idea of directly learning to approximate the Shapley value is interesting, it is unclear from the paper how the module can learn to approximate Shapley interaction and estimate its uncertainty. Specifically, what is the input of this module? Is it the set of all visual+textual tokens? Is the uncertainty estimation reliable as neural networks are notorious for being very confident in their predictions?

---

> ### Author Response · Authors · 2022-08-02
> **Response to Reviewer yCwd (Part 1)**
>
> We sincerely thank you for the valuable comments. We are encouraged to see that our work is recognized as novel and interesting. We will explain your concerns point by point.
>
> **Q1: The technical section could be challenging to follow, especially for computer vision audiences without prior knowledge of Shapley value. The reviewer believes it would be very helpful if the paper could include a table to highlight key differences of the proposed work with various methods used for vision-language pretraining. If the reviewer understands correctly, the main distinction compared to other works would be the proposal of using Shapley interaction as soft pseudo labels for fine-grained image regions instead of using contrastive learning between image-caption pairs. However, this is not clear and emphasized enough in the paper.**
>
> **A1:** Thanks for your nice suggestion. We highlight key differences in the following table. As you correctly understand, our LOUPE differs as it explicitly learns fine-grained region-phrase alignment from the novel perspective of game-theoretic interactions, without resorting to any object-level human annotations or pre-trained Region Proposal Network (RPN). Notably, the human bounding-box annotations are usually limited to the pre-defined object categories, and the RPN can only detect regions belonging to the pre-defined categories of pre-training object detection datasets. Thus, the methods that use human bounding-box annotations or pre-trained RPN usually suffer from detecting novel objects beyond the pre-defined categories. In contrast, our LOUPE learns from large-scale raw image-text pairs, which are more scalable and contain a broader set of visual concepts. For example, as shown in the right case of Figure 4, LOUPE successfully recognizes the leash region and aligns it with the “a leash” phrase. Note that the “leash” category has never appeared in any existing object detection datasets. We will include these analyses in the next version according to your valuable suggestion.
>
> | Methods                               | Coarse-grained image-text alignment | Fine-grained region-phrase alignment | Ways to learn fine-grained region-phrase alignment |
> | :------------------------------------ | :---------------------------------: | :----------------------------------: | :--------------------------------------: |
> | CLIP, ALIGN, DeCLIP                   |            $ \checkmark$            |                  -                   |                    -                     |
> | ImageBERT, UNITER, FILIP, ViLT, ALBEF |            $ \checkmark$            |            $ \checkmark$             | Implicit supversion signals from end-to-end training (*e.g.,* Image-Text Contrastive loss) |
> | GLIP, X-VLM, RegionCLIP               |            $ \checkmark$            |            $ \checkmark$             | Human bounding-box annotations or supervised pre-trained Region Proposal Network |
> | **LOUPE**                             |            $ \checkmark$            |            $ \checkmark$             | Explicit alignment information quantified by game-theoretic interactions |
>
> **Q2: The reviewer is unclear how Shapley value could be used as soft pseudo labels. For example, in line 180, the paper mentions that the game score v1, which is used to compute the Shapley interaction, is the similarity between image and text. However, it appears that the paper does not mention any training phase for learning image-textual similarity. Then how would this similarity be computed correctly? Are there any pretraining phases or pretrained models employed for similarity computation?**
>
> **A2:** We appreciate the reviewer's concern about the reliability of Shapley value. As you nicely point out, the reliability of Shapley value depends on the performance of computing image-text similarity. In practice, we first pre-train the image encoder and text encoder only based on the image-text contrastive loss in the first epoch and add Shapley interaction modeling in the remaining epochs. Also, the zero-shot transfer performances on object detection and visual grounding verify the reliability of Shapley value. We will clarify this in the next version.

---

> > ### Author Response · Authors · 2022-08-02
> > **Response to Reviewer yCwd (Part 2)**
> >
> > **Q3: The paper lacks convincing motivation on why using Shapley value instead of token-wise alignment (FILIP [42]) is beneficial. Although in lines 97-99, the paper mentions that FILIP has quadratic complexity, the proposed method also suffers from combinatorial complexity, which could be worse.**
> >
> > **A3:** Thanks for raising an important point. The superiorities of using Shapley Interaction modeling are mainly three-fold:
> >
> > 1. We suppose that directly computing token-wise alignment between every patch token and word token is not efficient and meaningful because an individual word token or patch token might not contain complete semantics. A semantic-rich phrase (*e.g.,* “a girl in a blue coat”) usually consists of multiple words, and its corresponding visual region is composed of multiple patches. Also, some words (*e.g.*, "is", "the") and patches (*e.g.*, background pixels) are not meaningful. Based on this insight, our LOUPE differs as we first propose token-level Shapley interaction modeling to aggregate patches into semantic-meaningful regions, and then introduce semantics-level Shapley interaction modeling to explicitly model the fine-grained semantic alignment between semantic-meaningful regions and phrases.
> > 2. Although FILIP computes token-wise similarity to simulate the fine-grained alignment, it can only learn implicit alignment from the indirect supervision of image-text contrastive loss, lacking training signals to explicitly encourage semantic alignment between visual regions and textual phrases. In contrast, our Shapley interaction modeling provides explicit supervision signals (*e.g.*, the alignment matrices visualized in Figure 4) to learn the fine-grained alignment. The consistently superior performance of our LOUPE than FILIP over all metrics also demonstrates the benefit of explicit fine-grained alignment learning.
> > 3. FILIP can not be directly applied to object detection and visual grounding through implicit token-wise alignment learning while our LOUPE can immediately transfer to these tasks without any fine-tuning. It is because the proposed Shapley interaction modeling enables our model to identify semantic regions and align these regions with language. As shown in Table 2, without any bounding-box annotations and fine-tuning, our LOUPE achieves competitive performance across four object detection and visual grounding benchmarks.
> >
> > Furthermore, FILIP has quadratic complexity with respect to the number of patch tokens and word tokens while we only compute the alignment between every region and phrase, and the number of regions and phrases is usually much less than the number of word and patch tokens. We will clarify this in the next version.
> >
> > **Q4: Although the idea of directly learning to approximate the Shapley value is interesting, it is unclear from the paper how the module can learn to approximate Shapley interaction and estimate its uncertainty. Specifically, what is the input of this module? Is it the set of all visual+textual tokens? Is the uncertainty estimation reliable as neural networks are notorious for being very confident in their predictions?**
> >
> > **A4:** Thanks for your question. We have implemented three versions of models to approximate Shapley interaction and they take all visual and textual tokens and the index corresponding to the target region or region-phrase pair as input. We respectfully refer the reviewer to Appendix D, where we provide the implementation details of these three versions.
> >
> > To investigate the reliability of the uncertainty estimation, we measure the correlation coefficient between the relative error and the uncertainty based on the results reported in Figure 2 (b) and (c). The correlation coefficient is a statistical measure to quantify the correlation degree between two variables. The values range between -1 and 1. A correlation of 1 indicates a strong positive correlation. Specifically, in Figure 2 (b) and (c), we test the estimation model on 1000 samples and report their mean uncertainty and relative error. The estimation model predicts a target Shapley interaction value and corresponding uncertainty $\sigma$ for each testing sample. We compute the relative error for each prediction according to the results computed by the sampling-based method. Then, we measure the correlation coefficient between the relative error and the uncertainty. For token-level Shapley interaction, the average correlation coefficient is 0.74. For semantics-level Shapley interaction, the average correlation is 0.86. The results indicate that our Shapley interaction learning module tends to estimate higher uncertainty for testing samples with larger relative errors. Therefore, the uncertainty estimation is a reliable indicator, which helps us to determine whether to use the neural Shapley interaction learning module or the sampling-based method.

---

> > > ### Comment · Reviewer_yCwd · 2022-08-08
> > > **Response to rebuttal**
> > >
> > > Thank the authors for their response.
> > >
> > > The rebuttal answers/clarifies most of my questions as well as my concerns.
> > > I still strongly suggest that the methodology section could be simplified or re-organized to improve the paper's readability (as well as its impact).
> > >
> > > Overall, I recommend the paper for acceptance.

---

> > > > ### Author Response · Authors · 2022-08-09
> > > > **Thank you for your acknowledgement!**
> > > >
> > > > We really appreciate your precious time and valuable suggestions. As you nicely point out, we will carefully improve the organization and expression of the methodology section.

---

### Author Response · Authors · 2022-08-02
**General Response to All Reviewers**

We thank all the reviewers for their insightful and valuable comments! Overall, we are encouraged that they find that:

1. The idea of learning fine-grained region-phrase alignment from the perspective of game-theoretic interactions is **quite interesting and novel** **(all reviewers)**.
2. Viewing weakly-supervised region-phrase alignment in terms of game theory is a **promising direction to explore** (Reviewer yCwd).
3. It is **quite impressive** that our method can be used as a zero-shot object detector after pre-training on raw image-text pairs without bounding-box annotations (Reviewer gkzN).

We have revised the manuscript according to the reviewers' comments. The main changes we made include:

1. In Appendix I, we add experiments of zero-shot image classification over 11 datasets.
2. In Appendix J, we add experiments of linear probing over 11 datasets.
3. In Appendix K, we add a training cost-performance comparison table and discuss the training efficiency of our method.
4. In Appendix L, we add a comparison table to highlight key differences of our LOUPE with various methods. Also, we add a detailed discussion with some related works (*i.e.*, FILIP, RegionCLIP, X-VLM).
5. In Appendix E, we add more details about pre-training dataset construction.

Next, we address each reviewer's detailed concerns point by point. We hope we have addressed all of your concerns. Discussions are always open. Thank you!

---

### Meta-Review · Area_Chair_RmMc · 2022-08-25

**Recommendation:** Accept
**Confidence:** Certain

**Metareview:**

This paper addressed the fine-grained visual-language alignment from the perspective of game-theoretic interactions. It received diverse scores with three weak accept and one week reject. The technical novelty is acknowledged by all reviewers. The initial reviews raised concerns about unclear explanations of Shapley, insufficient experiments, and comparison. During the discussion, the authors have addressed most of the concerns by adding tables to compare with existing works,  adding zero-shot classification experiments in 11 datasets, and sufficient discussion with previous works. Reviewer H9ch mentions possible unfair comparison. The authors have reimplemented the CLIP with a fair encoder and the performances have been consistently improved. The meta-reviewers thus suggest accepting this paper. However, as highlighted by most reviewers, the method is too complex and hard to make readers easily follow, especially the Shapley part. It is strongly encouraged to revise the camera-ready paper. And the proposed 240M dataset is better released to facilitate future research to follow and compare, and explained about filter strategy and possible ethical issues, as addressed by two reviewers and promised by the authors.

**Award:**

No

---

### Decision · Program_Chairs · 2022-09-14

Accept